# Prospects for silvicultural enhancement of fire resistance in mesic westside forests of the Pacific Northwest

Sebastian U. Busby[1,2*], Jeremy S. Fried[1]

**1** United States Department of Agriculture Forest Service, Pacific Northwest Research Station, Portland Oregon, United States of America, **2** Oak Ridge Institute for Science and Education, Oak Ridge Tennessee, United States

\* sebastian.busby@tnc.org

## Abstract

Increasing wildfire activity in mesic, temperate Pacific Northwest forests west of the Cascade Range crest has stimulated interest in understanding whether alternative forest management practices could reduce risk of stand-replacing fire. To explore how management can enhance fire resistance in these forests and assess tradeoffs among resistance enhancement, carbon sequestration and storage, and economic returns, we conducted 40-year simulations of stand development with BioSum, a framework for conducting landscape analysis with the Forest Vegetation Simulator (FVS), utilizing a statistically representative and spatially balanced sample of Forest Inventory and Analysis (FIA) plots. Simulation outcomes under business-as-usual silviculture were contrasted with fire-aware silviculture, and treatment optimization logic was developed and applied to represent landscape-scale outcomes under business-as-usual and fire-focused management scenarios. Simulation results indicate that fire-aware prescriptions and fire-focused management can meaningfully enhance stand- and landscape-scale fire resistance of westside forests under less than extreme fire weather, but at the cost of lower economic returns and reduced net carbon storage and sequestration over the 40-year analysis window. Shifting from business-as-usual regeneration harvests with short rotations to fire-aware, episodic selection harvest improved fire resistance the most, especially in young privately-owned forests, and with only modest tradeoffs in carbon and economic outcomes. While fire-aware treatments generally reduced net present value from forest oper-ations over business-as-usual, most treatments still generated positive net present value and could be implemented without subsidy. Fire-aware prescriptions that removed and utilized non-merchantable harvest residues instead of burning them, via either pile or broadcast burning, partially mitigated carbon emissions associated with fire-aware treatments, with about the same improvement in fire resistance. Given the currently limited institutional and financial capacity to implement fire resistance

**Data availability statement:** The data supporting the findings of this study are available from the USDA Forest Service Forest Inventory and Analysis (FIA) program, but restrictions apply. All but exact plot location are downloadable (https://apps.fs.usda.gov/fia/datamart/datamart.html); exact plot locations are not publicly available owing to confidentiality requirements under the Food Security Act. The silvicultural prescription Forest Vegetation Simulator (FVS) keyword files crafted and simulated in this study, among other supporting data, are made publicly available in a Figshare repository (https://doi.org/10.6084/m9.figshare.26876341).

**Funding:** This research was supported by the U.S. Department of Agriculture (USDA) Forest Service PNW Research Station's Westside Fire Research Initiative, which funded appointments to the Research Participation Program administered by the Oak Ridge Institute for Science and Education (ORISE) through an interagency agreement #21IA11261979021 with the U.S. Department of Energy 840 (DOE). ORISE is managed by ORAU under DOE contract number DE-SC0014664. The findings and conclusions in this paper are the responsibility of the authors and should not be construed to represent any official view, determination or policy of the USDA, DOE, ORAU/ORISE or any other U.S. Government entity. The funders had no role in study design, data collection and analysis, decision to publish, or preparation of the manuscript.

**Competing interests:** The authors have declared that no competing interests exist.

enhancing treatments at scale, the insights from this analysis may aid managers seeking to elevate fire resistance to prioritize where and how to manage.

## Introduction

Over 8.5 million hectares of mesic, temperate, coniferous forests grow west of the Cascade Range crest in Washington and Oregon states, USA. These "westside" forests cover a complex matrix of ownership and management legacies and objectives, serving myriad ecological functions and providing a broad spectrum of ecosystem services that include habitat, timber production, recreation and tourism, and carbon storage and sequestration. Westside forests have developed under moist, cool, and productive climate conditions and a strong legacy of fire suppression, which have limited their exposure to wildfire over the past century and promoted forest structures with high fuel loads and horizontal and vertical fuel connectivity [1]. As climate warming continues to increase these forests' exposure to wildfire (i.e., longer fire seasons and greater seasonal forest fuel aridity, on average), substantial uncertainty persists surrounding what role, if any, active forest management can play in mitigating the spread and severity of future wildfires in the region.

Westside forests are increasingly recognized as exhibiting a complex fire regime comprised of three broad patterns: (1) infrequent wind-driven, very large, stand-replacing fires, (2) mixed severity fires of smaller spatial extent at intermediate frequency, that primarily burn under mild to moderate fire weather conditions, and (3) frequent short-interval fires in early seral forests created by stand-replacing disturbance (i.e., wildfire and/or logging [1–4]). Excepting two very large, wind-driven, stand-replacing fires and subsequent, short-interval reburns (the 1902 Yacolt and 1933–1951 Tillamook fires), this region experienced few large fires over the 20th century, likely owing to a relatively cool and wet climate (i.e., fire activity is climate-limited [5]) and active fire-suppression [6–7]. Coinciding with anthropogenic climate warming, fire activity in the region has been increasing since the late 20th century [8] and is projected to increase substantially by mid-to-late 21st century [9–10]. All three broad fire patterns described above have been observed during the early 21st century, including the wind-driven, stand-replacing Labor Day 2020 fires that burned over 300,000 hectares of forestland from southwestern Washington to central Oregon in the Cascade Range [11–13].

Under less-than-extreme fire weather (defined here as the combination of very dry fuels and high winds), forest structure can strongly influence fire behavior and tree mortality in mesic westside Pacific Northwest forests [1,13,14]. The typically high fine down wood loadings in these forests ensure surface fire propagation and can generate surface flame lengths sufficient to kill trees by multiple pathways: 1) crown scorch (killing all or most live foliage, without igniting it), 2) lethal cambium temperatures and/or root kill, 3) initiating passive crown fire by igniting lower branches in the forest canopy, which carry fire into the upper canopy, consuming foliage and fine branches, or 4) sustaining an active crown fire behavior under which fire spreads from crown

to crown [15]. Low stand-scale canopy base heights in forests dominated by or with significant stocking in young trees increases fire-induced tree mortality, because under even moderate fire weather (e.g., dry fuels and low winds), fire can ignite the lower branches of those young trees and propagate into their upper crowns, and beyond into the crowns of larger, older trees, if those are also present [16]. Thus, fire-induced tree mortality is commonly the greatest in stands with high surface fuel loading and high vertical and horizontal fuel connectivity [17].

Westside forest structure and composition have been significantly influenced and altered by European colonization and the displacement of native peoples since the 19th century [6,18]. Large scale logging efforts over the late-19th to mid-20th centuries significantly reduced old-growth forest extent and increased species dominance by Douglas-fir (*Pseudotsuga menziesii*) – a favored timber species. Persistent suppression of both lightning- and human-ignited wildfires, including intentional indigenous burning and unintentional fires ignited by European colonizers, are thought to have influenced forest structure and extent across the region post-colonization [4,18], albeit to a lesser degree than more xeric coniferous forests elsewhere [19–20], where fire activity is more fuel-limited.

Contemporary forest structure in the region derives from the forest management legacies and policies associated with different ownership groups. The National Forest System (NFS) oversees the largest public trust of forest area (~2.95m ha), followed by The Bureau of Land Management (BLM; ~1.00m ha), Washington Department of Natural Resources (~0.75m ha), and Oregon Department of Forestry (~0.30m ha). Active forest management on NFS lands has declined precipitously since the 1970's, in part due to constraints spawned by the 1973 Endangered Species Act, 1994 Northwest Forest Plan, and forest conservation movements of the era [21]. Forest Inventory and Analysis (FIA) data indicate that active management (thinning, clearcutting, and other silvicultural activities excluding incidental cutting) has occurred in recent history (past ~30 years) on less than 24% of unreserved westside NFS forestland. While active management for timber resources does occur on a subset of NFS lands, hazard tree removal (e.g., along roadways, powerlines, campgrounds, and recreational areas) is perhaps the most commonly observed active management action of the agency at a landscape scale. NFS lands contain the greatest share of mature and old-growth forests in the region, supported largely by a passive management strategy. BLM and state ownerships answer to a broader set of goals concerning timber production and provision of recreation and habitat. Forest age ranges widely but trends younger on BLM than on NFS forestlands. Mean stand age is even younger on State forests. Private, corporately owned forest is second in areal extent only to NFS (~2.70m ha). Private, non-corporate ownerships represent a smaller, but still significant, share of the forest, mostly comprised of family holdings but also including tribes, clubs and associations and non-governmental organizations such as land trusts (~0.79m ha). Private corporate forests are managed primarily for timber production and most are young, even-aged plantations of Douglas-fir (sometimes mixed with western hemlock [*Tsuga heterophylla*]), managed using clearcut harvests over rotations of short to moderate length (~35 to, at most, 60 years). Pile burning of slash commonly follows clearcut activity to facilitate planting and reduce fuel loading elevated by harvest operations.

To date, little to no active forest management in the region has focused on enhancing resistance to stand-replacing fire (i.e., to reduce likelihood of crown-fire and of high rates of tree mortality), referred to hereafter as fire resistance. On most forest area, silvicultural prescriptions applied in recent decades, including the "grow-only" prescription implied when no active management occurs, are intended to fulfill one of two objectives: 1) maximize net economic return (typical on corporate, some non-corporate, and some State and BLM forests) or 2) accelerate the development of old-growth structural characteristics (typical on NFS and common on other public ownerships). A few key factors may explain the absence of fire resistance enhancement from current management practice on all ownerships. Until quite recently, a dearth of large stand-replacing fire incidents has led landowners to perceive wildfire risk as low—a perception potentially reinforced by scientific reports and anecdotal narratives that label this region as an "asbestos forest" in which wildfire primarily acts as a rare, unpredictable, and catastrophic natural disturbance [1,22–24]. Landowners and managers may have selectively seized on the "rare" and "unpredictable" as a permission structure for discounting the "catastrophic". Moreover, it is close to conventional wisdom among these westside forest decisionmakers that, unlike dry forests, mitigation of fire effects via

fuel treatment is impractical due to abundant fuels that necessitate frequent and costly re-treatment to sustain enhanced resistance. Such beliefs appear to have had staying power, notwithstanding a near absence of supporting evidence.

Extensive experimentation has been performed and synthesized to assess fuel treatment efficacy and capacity to reduce crown fire across pine and mixed conifer forests throughout the Western U.S. - excepting westside PNW forests, consistently producing findings that combinations of thinning and surface fuels reduction via broadcast and pile burning do confer some degree of resistance to crown fire/stand replacing fire [25–27], particularly when using low (thin-from-below) thinning that addresses ladder fuels, not just canopy bulk density reduction [28]. When surface fuel treatments are not an option, thinning with whole tree harvest can still elevate resistance via ladder fuel reduction that avoids generating activity fuels, and [29] reported that longevity of fuel treatment effects can be extended by re-treatment. None of those studies suggest that fuels treatments can't work in westside forests, but we found no literature reporting on empirical experiments designed to test the potential for silvicultural practice to sustainably enhance stand-scale fire resistance, or on the costs and effectiveness of fuel treatment in westside forests [30]. Fire suppression continues to be offered as the primary adaptation strategy to climate warming in historically cool and mesic forests [7,31], despite these forest's increasing wildfire risk (i.e., non-stationary climate [32]), an ongoing national-scale wildfire crisis [33] and inadequate fire suppression capacity (e.g., acute shortages of trained personnel).

The adverse social, ecological, and economic impacts that may result from an increasing incidence of large stand-replacing fires in westside forests make a compelling case for improving our understanding of how active forest management might reduce such impacts. Making changes to silvicultural regimes to enhance resistance will also affect other widely sought services from forests, including economic returns and carbon storage and sequestration. To address these questions at stand- and landscape-scales, we utilized (1) a forest growth and yield simulation model (FVS; Forest Vegetation Simulator), (2) a silvicultural treatment cost and effectiveness modeling framework (BioSum; Bioregional Inventory Originated Simulation Under Management), (3) a statistically representative and spatially balanced sample of westside forests (from the USDA Forest Service Forest Inventory and Analysis [FIA] program), and (4) surveys and transcripts of interviews with owners and managers of large, westside forest holdings, to simulate, compare, and contrast the outcomes of business-as-usual management with alternative, fire-focused management scenarios that deploy fire-aware silviculture on westside forests over the next 40-years. We address three research questions:

1. To what degree can fire-aware silvicultural treatments and fire-focused forest management improve stand- and landscape-scale fire resistance of westside forests relative to business-as-usual?

2. What tradeoffs can be expected in terms of economic returns and forest carbon storage and sequestration when managing to promote fire resistance?

3. How do fire resistance and carbon outcomes differ between fire-aware silvicultural treatments that utilize (i.e., remove and utilize) vs. burn (i.e., pile and/or broadcast burn) harvest residues?

## Materials and methods

### Study area

Our study area encompassed productive, conifer-dominated forests found west of the Cascade Range crest in Oregon and Washington states of the US Pacific Northwest region (Fig 1), in two mountainous ecoregions: the western Cascades and Coast Range. Climate is mesic Mediterranean with cold, wet winters and warm, dry summers. East of the Coast Range crest, mean annual precipitation increases and temperature decreases with increasing latitude and elevation, thus, low-elevation and -latitude forests are markedly warmer and drier than high-elevation and -latitude forests, strongly influencing forest productivity, density, and species composition [34]. Low-elevation forests in the western Cascades and much of the Coast Range forests are privately owned, managed for timber production, and dominated by young Douglas-fir

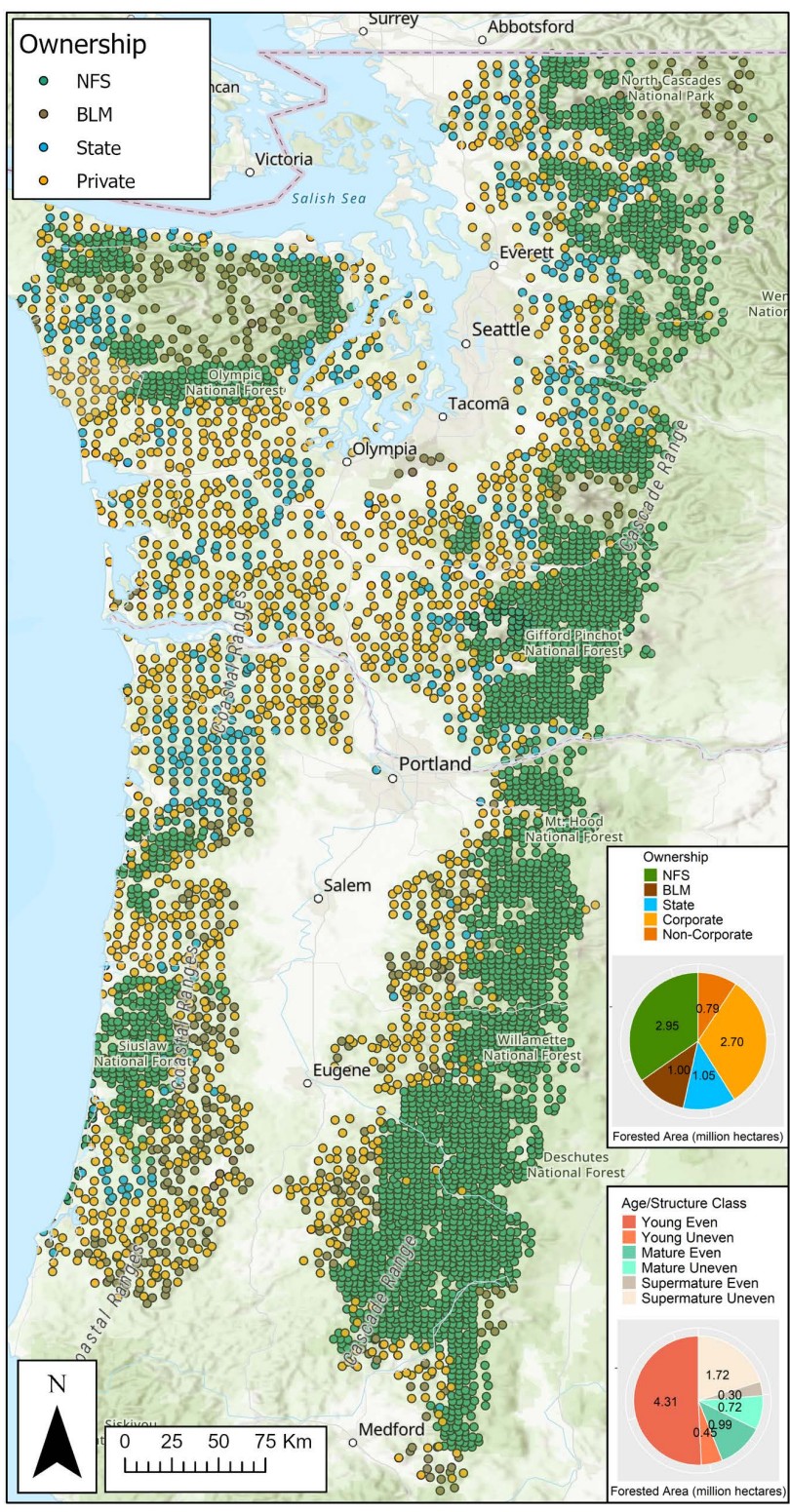

**Fig 1. Approximate locations of forested Forest Inventory and Analysis (FIA) plots modeled to represent management trajectories in westside Pacific Northwest forests in Oregon and Washington states, USA.** Inset graphs summarize forested area (hectares) by ownership and age/structure class as represented by these plots.

(*Pseudotsuga menziesii)* plantations, except west of the Coast Range crest where western hemlock (*Tsuga heterophylla*) and Sitka spruce (*Picea sitchensis*) better tolerate exceedingly high precipitation and dominate or contribute a major share of tree biomass. A few large State (e.g., Tillamook and Clatsop) and National Forests (Siuslaw) cover parts of the Coast Range in Oregon. Mid-to-high elevation forests in the western Cascades are largely administered by the USDA National Forest Service as part of the national forest system (NFS) or by the Bureau of Land Management (BLM) as district forests. A large proportion of high-elevation forests in the Coast Range and western Cascades are within National Parks (e.g., Crater Lake, Mt. Rainer, Olympic, and North Cascades) or designated as wilderness within national forests, where no active management occurs. At mid-elevations, tree species composition is commonly dominated by western hemlock and/or Douglas-fir, and at high-elevations by Pacific silver fir (*Abies amabilis*), noble fir (*Abies procera*), or mountain hemlock (*Tsuga mertensiana*). Hardwood and pine species generally comprise a minor component across the region, except under the warmest and driest climate conditions found in southern Oregon. The fire resistance of dominant tree species in the region is considered relatively low (i.e., thin barked, high crown ratios), although mature (e.g., DBH > 25 cm) Douglas-fir can exhibit moderate-to-high resistance to surface fires [35].

## Forest inventory data

We used data from a statistically representative and spatially balanced sample of field-measured Forest Inventory and Analysis (FIA) plots, from the most recently available evaluation identifier (with panels assessed in 2008–2019), as inputs to forest growth and yield model simulations in the Forest Vegetation Simulator (FVS). Sampled area within each FIA plot is partitioned into separate "conditions" when delineation-qualifying differences (e.g., in owner group, forested status, reserve status, forest type, stand size class, tree density class, topographic features) exist and minimum condition size and shape criteria are satisfied. Site attributes such as slope, site quality, stand age, forest type and landowner group are defined and collected at the condition level, and each condition can be thought of as a forest "stand" that represents a statistically based sample corresponding to a definable number of forested acres within a broader landscape. We used FIA conditions, hereafter referred to as stands, as the basic simulation and analysis unit in this study. Further, we used the Bioregional Inventory Originated Simulation Under Management (BioSum) analysis framework and workflow management software [36] to prepare and pre-process FIA stand data for simulation in FVS and to generate and analyze economic data associated with FVS simulation output; 6,396 forested stands were retained for simulation (Fig 1). For additional methodological context regarding FIA plot selection and data preprocessing methods, see the Forest inventory data section of the supplementary file (S1 Appendix).

## Silvicultural treatments

To represent and simulate business-as-usual (BAU) forest management practices in westside forests, we constructed silvicultural prescriptions (treatments; Rx) based on data obtained from survey and interview responses of large forest landowners in the study area (a stratified random sample of Private Corporate, Federal, State, and Tribal groups), to support the USDA Forest Service Pacific Northwest Research Station's Westside Fire Research Initiative. Survey and interview questions asked specifically about common agency-level forest management practices (e.g., harvest systems, rotation lengths, commercial and non-commercial thinning parameters, species retention and planting preferences, thinning and harvest triggers and targets). Formal ethics approval for this study was not obtained because all researchers facilitating surveys and interviews were U.S. Forest Service scientists. The Forest Service has no ethics approval or Institutional Review Board (IRB) process but does have a scientific integrity policy that was followed. For additional methodological context on how interview and survey data were collected, processed, and interpreted, see the Silvicultural treatments section of the supplementary file (S1 Appendix).

BAU treatments were associated with representative examples of what can be loosely described as even- or uneven-aged management styles, differentiated by the former ultimately involving a stand-replacing harvest (e.g., clearcut and replant with potential pre-commercial or commercial thinning entry) and the latter a sequence of individual tree selection

harvests, with trees thinned evenly across diameter classes at each entry, to maintain recruitment of new, younger growing stock over time via natural regeneration. To contrast BAU management and treatments, which largely focus on maximizing stand growth or enhancing old growth structural characteristics, we developed a suite of "fire-aware" even- and uneven-aged treatments aimed at enhancing stand-scale fire resistance. Fire-aware treatments are designed to enhance fire resistance over BAU by 1) lowering tree density triggers for commercial thinning entries and thinning to lower residual tree densities, 2) thinning from below and replanting at lower densities for even-aged treatments, and 3) reducing surface fuel loading and ladder fuels by thinning ladder fuels (non-merchantable trees < 15 cm diameter) and removing residues, pile burning residues, or broadcast burning.

Finally, to ensure our BAU and fire-aware silvicultural prescriptions were realistic, we elicited feedback from regional silviculturists and revised, removed, or added prescriptions as recommended. Once finalized, each silvicultural prescription was coded as an FVS keyword control parameters file, to control simulation in FVS. We tested each prescription file in FVS on a subset of stands and reviewed the simulation outputs to verify that silvicultural activities were being applied as intended, as a quality assurance step in advance of running simulations across the whole landscape. See Table 1 for the full list of the silvicultural prescriptions developed and simulated in this study.

## Growth and yield simulations

Each prescription was simulated for 40 years in FVS for each stand, along with a grow-only (i.e., no management) prescription. 40-years is the outer limit for which FVS is considered plausibly reliable for management support decision making, given that its equations are fit to inventory remeasurement series on the order of 1–2 decades [37]. We used a deterministic modeling approach in FVS, where a single simulation was processed per scenario. To account for wildfire, as described under the "Wildfire activity and prevalence" later in this section, four alternative iterations of these simulations for each prescription and stand included the SIMFIRE keyword from the Fire and Fuels Extension (FFE) to FVS with parameters representing mild-moderate or severe fire weather conditions, under two timings of fire occurrence. The mild-moderate and severe fire weather scenarios were parameterized to FFE variant-level default fuel moisture and temperature settings, with average wind speed reduced to 2.2 and 6.7 m/s (vs. FFE defaults of 2.7 and 8.9 m/s). These reduced wind speed settings better represent weather under which fires burn, with mild-moderate parameters applicable to most wildfire events (which account for a small burned area proportion) and severe parameters consistent with historically less frequent conditions linked to severe wildfires (which already account for a majority of burned area, with potentially an even larger share under climate warming). Weighted pairs of the original 13 fuel models were assigned by FFE, as guided by stand-level FIA measurements of down wood and litter, rather than relying on field crew observed surface fuel models, given related analysis of the 2020 Labor Day fires found FFE-predicted fire effects based on a single crew-called model to be less accurate [13].

To achieve synchrony between FVS outputs and the BioSum modeling framework, we simulated four, 10-year BioSum "cycles" in which management could occur (if triggered by the prescription) at the beginning of each cycle (FVS simulation years 2, 12, 22, and 32, corresponding to BioSum cycles 1, 2, 3, and 4). Owing to FVS output from FFE containing only post-treatment values in a treatment year and the requirement when analyzing a silvicultural trajectory for both pre- and post-treatment stand attributes at every stand entry, we set projection timings in FVS to generate output for simulation years 1, 2, 3, 11, 12, 13, 21, 22, 23, 31, 32, and 41, using years ending in 1 for pre-treatment values in FFE-generated tables, ending in 2 for pre-treatment values in FVS-generated tables, and ending in 3 to obtain post-treatment attributes from all tables - with the exception of 41, which provides a more "settled" outcome of the final treatment at year 32.

## Natural tree regeneration

While FVS can model regeneration endogenously in variants covering western Montana, central and northern Idaho and coastal Alaska, users of all other variants, including the two that apply in our study area, can only account for regeneration

**Table 1. Business-as-usual (BAU) and fire-aware (FA) even-aged (EA; regeneration harvest) and uneven-aged (UA; selection harvest) silvicultural prescriptions (Rx) developed in this study and their descriptive parameters.**

| Rx ID | Harvest System | Species Retain | Pre-Commercial Thin | Commercial Thin | Rotation Harvest | Fuel Treatment |
|---|---|---|---|---|---|---|
| BAU-EA1 | Whole Tree | WRC, WH, WP > DF | Trigger: >740 TPH & Age < 20; Target: 556 TPH | Trigger: Age 35-60 & SDI > 55% SDI-MAX & MCM > 12; Target: Thin evenly across diameter to 30% SDIMAX trees 15-76cm DBH | Trigger: Age > 50 & CMAI; Target: Clearcut, apply herbicide, replant 618 TPH (432DF/62WH/62WRC/62WP) | NA |
| BAU-EA2 | Log Length | WRC, WH, WP > DF | Trigger: >740 TPH & Age < 20; Target: 556 TPH | Trigger: Age 35-60 & SDI > 55% SDI-MAX & MCM > 12; Target: Thin evenly across diameter to 30% SDIMAX trees 15-76cm DBH | Trigger: Age > 50 & CMAI; Target: Clearcut, apply herbicide, replant 618 TPH (432DF/62WH/62WRC/62WP) | NA |
| BAU-EA3 | Whole Tree | WRC > WH > DF | Trigger: >740 TPH & Age < 20; Target: 556 TPH | Trigger: Age 35-60 & SDI > 55% SDI-MAX & MCM > 12; Target: Thin evenly across diameter to 30% SDIMAX trees 15-76cm DBH | Trigger: Age > 50 & CMAI; Target: Clearcut, apply herbicide, replant 618 TPH (246DF/246WH/126WRC) | NA |
| BAU-EA4 | Log Length | WRC > WH > DF | Trigger: >740 TPH & Age < 20; Target: 556 TPH | Trigger: Age 35-60 & SDI > 55% SDI-MAX & MCM > 12; Target: Thin evenly across diameter to 30% SDIMAX trees 15-76cm DBH | Trigger: Age > 50 & CMAI; Target: Clearcut, apply herbicide, replant 618 TPH (246DF/246WH/126WRC) | NA |
| BAU-EA5 | Whole Tree | WRC > DF > other spp. | NA | Trigger: Age 25-35 & MCM > 12; Target: Thin evenly across diameter to 35% SDIMAX trees > 15cm DBH | Trigger: Age > 50; Target: Clearcut, apply herbicide, replant 740 TPH (740DF) | NA |
| BAU-EA6 | Whole Tree | WRC > DF > other spp. | Trigger: Age 10-20; Target: 680 TPH | Trigger: Age 25-35 & MCM > 12; Target: Thin evenly across diameter to 35% SDIMAX trees > 15cm DBH | Trigger: Age > 50; Target: Clearcut, apply herbicide, replant 740 TPH (740DF) | NA |
| BAU-EA7 | Whole Tree | WRC > DF > other spp. | Trigger: Age 10-20; Target: 680 TPH | NA | Trigger: Age > 50; Target: Clearcut, apply herbicide, replant 740 TPH (740DF) | NA |
| BAU-EA8 | Whole Tree | WRC > DF > other spp. | NA | NA | Trigger: Age > 35; Target: Clearcut, apply herbicide, replant 740 TPH (740DF) | NA |
| BAU-EA9 | Whole Tree | WRC > other spp. | NA | NA | Trigger: Age > 35; Target: Clearcut, apply herbicide, replant 740 TPH (310DF/310WH/120WRC) | NA |
| BAU-UA1 | Whole Tree | WRC, WH, WP > DF | NA | Trigger: SDI > 55% SDIMAX & MCM > 12 & Years Since Treatment > 20; Target: Thin evenly across diameter to 30% SDIMAX trees 15-76cm DBH | NA | NA |
| BAU-UA2 | Log Length | WRC, WH, WP > DF | NA | Trigger: SDI > 55% SDIMAX & MCM > 12 & Years Since Treatment > 20; Target: Thin evenly across diameter to 30% SDIMAX trees 15-76cm DBH | NA | NA |
| BAU-UA3 | Whole Tree | WRC > DF > other spp. | NA | Trigger: SDI > 60% SDIMAX & MCM > 12 & Years Since Treatment > 20; Target: Thin evenly across diameter to 35% SDIMAX trees > 15cm DBH | NA | NA |
| FA-EA1 | Whole Tree | WRC > DF> other spp. | NA | Trigger: Age 20-45 & SDI > 45% SDI-MAX; Target: Thin from below to 20% SDIMAX trees 2.5-76cm DBH | Trigger: Age > 50 & CMAI; Target: Clearcut, apply herbicide, replant 500 TPH (350DF/50WH/50WRC/50WP) | Thin ladder fuels and remove |
| FA-EA2 | Whole Tree | WRC > DF> other spp. | NA | Trigger: Age 20-45 & SDI > 45% SDI-MAX; Target: Thin from below to 20% SDIMAX trees 2.5-76cm DBH | Trigger: Age > 50 & CMAI; Target: Clearcut, apply herbicide, replant 500 TPH (350DF/50WH/50WRC/50WP) | Thin ladder fuels and pile burn |

*(Continued)*

**Table 1.** (Continued)

| Rx ID | Harvest System | Species Retain | Pre-Commercial Thin | Commercial Thin | Rotation Harvest | Fuel Treatment |
|---|---|---|---|---|---|---|
| FA-EA3 | Whole Tree | WRC > DF> other spp. | NA | NA | Trigger: Age > 35; Target: Clearcut, apply herbicide, replant 500 TPH (500DF) | NA |
| FA-EA4 | Whole Tree | WRC > DF> other spp. | NA | Trigger: Age 20-45 & SDI > 45% SDI-MAX; Target: Thin from below to 20% SDIMAX trees 2.5-76cm DBH | Trigger: Age > 50 & CMAI; Target: Clearcut, apply herbicide, replant 500 TPH (350DF/50WH/50WRC/50WP) | Thin ladder fuels and broadcast burn |
| FA-EA5 | Whole Tree | WRC > DF> other spp. | NA | Trigger: Age 20-45 & SDI > 45% SDI-MAX; Target: Thin from below to 20% SDIMAX trees 15-76cm DBH | Trigger: Age > 50 & CMAI; Target: Clearcut, apply herbicide, replant 500 TPH (350DF/50WH/50WRC/50WP) | NA |
| FA-UA1 | Whole Tree | Pine > DF> other spp. | NA | Trigger: SDI > 45% SDIMAX & Years since treatment > 20; Target: Thin evenly across diameter to 20% SDIMAX trees 2.5-76cm DBH | NA | Thin ladder fuels and remove |
| FA-UA2 | Whole Tree | Pine > DF> other spp. | NA | Trigger: SDI > 45% SDIMAX & Years since treatment > 20; Target: Thin evenly across diameter to 20% SDIMAX trees 2.5-76cm DBH | NA | Thin ladder fuels and pile burn |
| FA-UA3 | Whole Tree | Pine > DF> other spp. | NA | Trigger: SDI > 45% SDIMAX & Years since treatment > 20; Target: Thin evenly across diameter to 20% SDIMAX trees 2.5-76cm DBH | NA | Thin ladder fuels and broadcast burn |
| FA-UA4 | Whole Tree | Pine > DF> other spp. | NA | Trigger: SDI > 45% SDIMAX & Years since treatment > 20; Target: Thin evenly across diameter to 20% SDIMAX trees 15-76cm DBH | NA | NA |

**Acronyms:** DF = Douglas-fir; WH = western hemlock; WRC = western red cedar; WP = western white pine; SDI = Stand Density Index; SDIMAX = maximum SDI; TPH = trees per hectare; DBH = diameter at breast height; CMAI = culmination of mean annual increment; MCM = merchantable cubic meters.
**Trigger:** Stand conditions that must be true to apply the silvicultural prescription.

**Target:** The desired stand conditions achieved due to the application of the silvicultural prescription.

by specifying quantities and species of seedlings and saplings that will appear at each simulation time-step [38]. Silvicultural treatments that reduce canopy density expose the forest floor to increased sunlight, promoting pulses of natural tree establishment that can contribute to ladder fuels, ultimately offsetting gains in fire resistance from those treatments over time. We developed REGIMPUTE [39] and incorporated it into our FVS workflow as a variant-level, keyword add file crafted from empirical observations of natural tree regeneration from thousands of FIA plots, to dynamically impute regeneration based on a stand's structural stage.

### Treatment cost, revenue, and yield

Data from the FVS_CUTLIST (harvested trees) output table and additional harvest cost flags set in FVS_COMPUTE for costs not directly associated with harvest were loaded into BioSum's Processor module and its R-based OpCost model [40] to estimate treatment cost and revenues from sales of merchantable timber volume and forest residues recoverable for energy. Contemporary unit costs for surface fuel treatments, planting, and site prep (S1 Table), were added to the onsite operations costs estimated via OpCost if integral to a prescription or set as flags in FVS. Non-merchantable wood forwarded to the landing was piled and burned there (with carbon emissions tracked) or, if haul cost per green ton to the

nearest bioenergy facility didn't exceed the assumed delivered price of $30 US dollars, or under scenarios in which utilization was mandated, then chipped at the landing and trucked to that facility. BioSum used values in 2022 US dollars by species group and tree diameter range, estimated by a regional forest economist (Mike Buffo of Mason, Bruce & Girard) from log price data to estimate revenue from sales of merchantable timber (S2 Table). All costs and revenues were converted to present value using a discount rate of 4%.

## Wildfire activity and prevalence

To account for the impact of wildfire, including increases in area burned (i.e., due to more frequent severe weather and decline in successful suppression outcomes) on forest carbon, fire resistance, and economic outcomes over the simulation period, we developed a wildfire prevalence scenario from five realizations of wildfire activity simulations referenced in the "Growth and yield simulations" section, above, in which (for all but the no wildfire case) relied on the SIMFIRE FVS-FFE keyword to trigger fire in the year before a treatment year. The five simulations were thus: mild-moderate weather wildfire in year 11, mild-moderate weather wildfire year 31, severe weather wildfire in year 11, severe weather wildfire in year 31, and no wildfire. The wildfire prevalence scenario assumes 5% of forested area burns per decade over the 40-year simulation period, where 20% of area burns under mild-moderate and 80% under severe fire weather conditions, to reflect historical and contemporary observations that most area in the region burns under severe fire weather conditions. For each stand and prescription, we calculated an expected value for each simulation output, at each simulation timestep, as a weighted combination of the metrics generated by the five wildfire activity simulations (see S3 Table for weighting design). For example, if predictions associated with a simulation timestep across the five wildfire activity simulations are 5, 7, 9, 4 and 11, the weighted metric value corresponding to an assumed 5% area burned per decade wildfire prevalence scenario is (5*0.02) + (7*0.08) + (9*0.02) + (4*0.08) + (11*0.80) = 9.96.

## Simulation output metrics

As described in the next section, simulations rely on different optimization criteria depending on the scenario and landowner, each developed from simulated harvest (net present value [NPV], as generated wood value less harvest and haul costs), habitat quality (via the Old Growth Structure index [OGSI]) or fire resistance calculations, though these are also tracked and reported on even when not relied on as optimization criteria. We calculate two metrics of fire resistance as the predicted tree volume survival proportion (TVSP): 1) using parameters developed from the First Order Fire Effects Model (FOFEM [41]), which considers tree species, diameter class, and crown base height, and assuming surface flame lengths of 1.8-2.4m, and 2) by the Fire and Fuels Extension (FFE [42]) of FVS under severe fire weather conditions. FOFEM-TVSP does not rely on surface fuels characterization or predictions of flame length (value is user-supplied) that such fuels might support, focusing instead on a range of flame lengths for which treatments stand a high chance of having influence on whether fire becomes stand-replacing [43]. FFE-TVSP is linked to a particular weather scenario (temperature, fuel aridity, and wind speed) and is strongly dependent on FFE's surface fuel characterization, which can be quite different than observed surface fuel loading.

   To summarize these optimization criteria and other key simulation outcomes over the 40-year simulation period, we calculated a weighted average (0.125 weight applied to simulation output for years 3, 12, 13, 22, 23, 32, where years ending in 3 are post- and in 2 are pre-treatment, and 0.25 weight applied to year 41, because it "stands-in" for the final decade following the last potential treatment activity in year 32), change (year 41 – year 2 value), or sum (sum of values across simulation years) per output metric, as appropriate. We calculated (1) 40-year weighted averages for OGSI and TVSP, (2) 40-year change for live and dead wood carbon metrics, and (3) 40-year sums for harvested carbon, carbon emitted from wildfire and burning activities, and economic metrics like NPV. All calculations were completed in R [44]; see Table 2 for a full list of the processed metrics.

**Table 2. Stand-scale metrics generated and evaluated in this study, as derived from Forest Vegetation Simulator (FVS) simulation outputs directly or indirectly, including their 40-year calculation method, unit, and computation source(s).**

| Type | Metric | Calculation | Unit | Computation Source |
|---|---|---|---|---|
| Habitat* | Old growth structure index (OGSI), an index of habitat quality derived from large snag and live tree density and coarse woody debris cover. | 40-year weighted average | bounded (0.0 - 1.0) | [33] and FVS |
| Fire Resistance* | Tree volume survival proportion predicted by the First Order Fire Effects Model (FOFEM) for fires having surface flame length of 1.8-2.4m. | 40-year weighted average | bounded (0.0 - 1.0) | [34], FOFEM and FVS |
| Fire Resistance | Tree volume survival proportion predicted by Fire and Fuels Extension (FFE) under severe fire weather conditions: very dry fuels, 21°C average temperature, 6.7 m/s average wind speed. | 40-year weighted average | bounded (0.0 - 1.0) | FVS |
| Carbon[1] | Net carbon sequestration calculated as: $\Delta$ Live Wood + $\Delta$ Dead Wood + (Merchantable Harvest*0.6) + (Non-Merchantable Harvest*0.4[if utilized]) – Fire-Related Emissions. | 40-year change (live or dead wood) or sum | Mt C02e/ hectare | FVS |
| Carbon | Change in aboveground carbon storage calculated as: $\Delta$ Live Wood + $\Delta$ Dead Wood | 40-year change | Mt C02e/ hectare | FVS |
| Economic | Unreserved forested area receiving one or more treatment (stand entry). | 40-year sum | hectares | FVS and BioSum |
| Economic* | Net present value discounted at 4%. | 40-year sum | 2022 USD ($) | FVS and BioSum |
| Economic | Merchantable harvested wood volume (stems > 15 cm diameter). | 40-year sum | m³ | FVS and BioSum |
| Economic | Non-merchantable harvested wood volume (stems < 15 cm diameter). | 40-year sum | m³ | FVS and BioSum |

[1]Assumes 60% of carbon associated with merchantable harvest volume is sequestered into long-lived (>100 years) wood products and 40% of carbon associated with utilized (chipped, transported, and sold) non-merchantable harvest volume provides a fossil fuel carbon substitution benefit.

* Metric was used in treatment optimization criteria for one or more scenarios and/or ownerships.

## Management scenarios and treatment optimization

To represent and summarize stand- and landscape-scale fire resistance, carbon, and economic outcomes under business-as-usual (BAU) and alternative fire-focused (FF) and fire-focused carbon-aware (FFCA) forest management scenarios, we developed optimization logic for each scenario that assigns a single, "best" silviculture prescription to each forested stand, generating simulation results (i.e., computed 40-year metrics) associated with the implementation of that best prescription. This logic incorporated an assortment of FIA condition codes to identify initial stand attributes and determine whether stands were eligible for silvicultural treatment, and if so, what type of treatments (e.g., even- vs. uneven-aged) based on landowner-specific management objectives and constraints. All stands with a reserved status code (active management prohibited) were assigned a "grow-only" treatment in every management scenario. Active or grow-only prescriptions were assigned to stands without reserved status according to the logic coded for each forest management scenario:

BAU: Condition-level treatment codes were assessed for evidence of active management (e.g., thinning, regeneration harvest, planting, site prep, fuel treatments, but not incidental or firewood cutting) over the 60 years prior to the most recent inventory visit. Stands associated with public ownerships (NFS, BLM, and State) that did not have evidence of management were assigned no treatment; these can be thought of as stands that have fallen out of (or never received) active management. All unreserved stands on Private ownerships (including corporate and non-corporate entities) were eligible for treatment regardless of their management history. Across ownerships, the initial stand age/structure classification observed at the most recent inventory visit determined whether a stand could receive even- or uneven-aged treatments (i.e., clearcut vs. selection harvest). Supermature (>120yrs) stands were only eligible for uneven-aged treatments, given their low likelihood of being clearcut. BAU prescriptions derived from conversations with public or private landowners could only be applied to forests in corresponding owner groups. Across treated stands, non-merchantable

harvest residues were utilized (chipped, transported, and sold) if their sell value was greater than their haul cost. Otherwise, they were pile burned at the operation landing (and carbon emission tracked). Treatment optimization criteria across ownerships and initial age/structure class involved either maximizing 40-year weighted average values of the Old Growth Structure Index (OGSI) or maximizing the 40-year sum of net present value (see S4 Table).

FF: Unreserved stands on all ownerships were (1) eligible for active management, (2) could receive BAU or fire-aware even- or uneven-aged treatments, and (3) non-merchantable harvest residues were always utilized regardless of whether the value obtained when delivered at the facility gate exceeded the costs of hauling them, to account for the possibility that new facilities might emerge to receive this material if fuels management were applied at scale. Treatment optimization criteria across ownerships and initial age/structure class were based on maximizing 40-year weighted average values of FOFEM-predicted tree volume survival proportion (see S4 Table).

FFCA: Treatment selection logic was identical to the FF scenario, except that fire-aware treatments that burned non-merchantable harvest residues through in-unit and/or landing pile burning or broadcast burning were ineligible. Thus, selection of the best prescription made under this scenario was restricted to FA_EA1, FA_EA3, FA_EA5, FA_UA1, or FA_UA4.

## Evaluating tradeoffs

To support evaluation of stand-scale tradeoffs between fire resistance (FOFEM-TVSP) and associated carbon (sequestration; change in aboveground storage) and economic (NPV; merchantable and non-merchantable harvest volumes) outcomes across management scenarios, we summarized area-weighted median and interquartile range statistics across outcome metrics by initial age/structure class and management scenario, irrespective of ownership for simplification. To identify if individual metric outcomes were statistically distinct between management scenarios among initial age/structure classes, we used Wilcoxon Rank Sum tests, appropriate for comparing groups with unequal sample sizes and non-parametric distributions, via the pairwise.wilcox.test() function in base R [44], with a Bonferroni correction for multiple pairwise comparisons, to test for statistical differences ($p < 0.05$) between management scenarios.

## Results

### Treatment optimization

Under the Business-As-Usual (BAU) management scenario, approximately 57% of unreserved forest area was actively managed during the 40-year simulation; uneven-aged stands on all ownerships were typically unmanaged, as were young and mature even-aged stands on NFS lands (Figs 2 and 3). Except for on NFS lands, most (~ 60%–90%) young, even-aged stands were actively managed. Management of mature even-aged forest was less common on State and BLM land (~ 30%–60%) than on Corporate and Non-Corporate (~ 90%–100%). Treatment effectiveness in even-aged stands on public lands (where objectives were tied to net present value or OGSI) was most frequently achieved via BAU_EA1; on private lands BAU_EA5 (~50yr rotation with potential commercial thin, replant 100% Douglas-fir) and BAU_EA9 (~35yr rotation, replant Douglas-fir, western hemlock, western redcedar mix) were most frequently selected to maximize net present value.

The scenario assumptions we specified led to assignment of active management prescriptions on about 84% of unreserved forest area under the FF management scenario and the proportion of uneven-aged stands managed was much greater than under BAU. Maximizing stand-scale fire resistance over the 40-year simulation period across ownerships resulted in fire-aware uneven-aged treatments (i.e., selection harvest) being allocated to most forest area, with FA_UA1 and FA_UA3 being the top two treatment choices. Both treatments thin across diameter to low tree densities (20% of MaxSDI) and cut non-merchantable ladder fuel trees (<15cm diameter), with FA_UA1 removing and utilizing those non-merchantable trees and FA_UA3 disposing of those felled trees via broadcast burn.

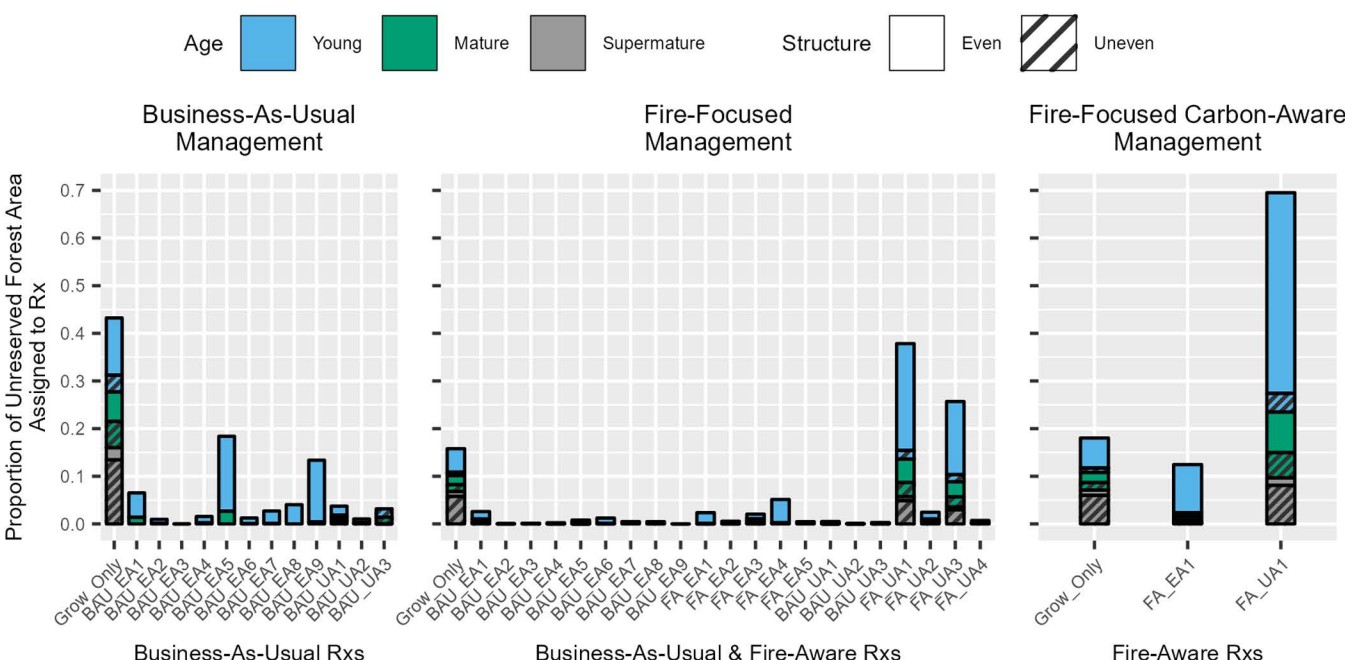

**Fig 2. The proportion of unreserved forest area, by initial age/structure class, assigned to each silvicultural treatment, under each of three management scenarios in which owner-specific objectives, that vary among owners and scenarios, are embedded.** Table 1 describes silvicultural treatment (Rx) parameters associated with labels on the x-axis.

Under the fire-focused carbon-aware (FFCA) management scenario, where treatments that include pile or broadcast burning were excluded from consideration to limit what could be considered anthropogenic carbon emissions from silvicultural activity, a similar degree of unreserved forest area was actively managed (~82%) as the FF scenario, with ~13% managed under FA_EA1 and the remainder (69%) under FA_UA1.

### Fire resistance

Under the BAU management scenario, FOFEM-predicted tree volume survival proportion (FOFEM-TSVP; fire resistance) at the stand-scale (area-weighted median value), was greatest across initial age/structure classes on BLM lands, followed by NFS, State, and Private lands (Fig 4). Across initial age/structure class and ownership, forests were generally more fire resistant in Oregon than in Washington. Mature even-aged forests were substantially more fire resistant on public lands (>75% median value) than private lands (~ 25% median value), while young, even-aged forests were the least fire resistant across ownerships (~ 30%–55% median value on public lands; ~ 20%–30% on private lands). Supermature stands were largely concentrated on public lands and exhibited high fire resistance (~ 50%–80% median value). At the landscape-scale, fire resistant forest area, defined as exhibiting >50% FOFEM-TVSP, was greatest across initial age/structure classes on BLM lands, followed by NFS and State lands, and was exceptionally low on private lands, which are largely dominated by young, even-aged forests (Fig 5).

At the stand-scale, FF and FFCA management scenarios typically enhanced fire resistance across initial age/structure class and ownership types relative to BAU, but to varying degrees. Fire resistance improvement under FF and FFCA relative to BAU was greatest in young even-aged forests across ownerships (~ 15%–35% increased median value) and young and mature even-aged forests on private lands (~ 30%–55% increased median value), and otherwise relatively modest across other age/structure class and ownership categories (~ 1%–20% increased median value). At the landscape-scale,

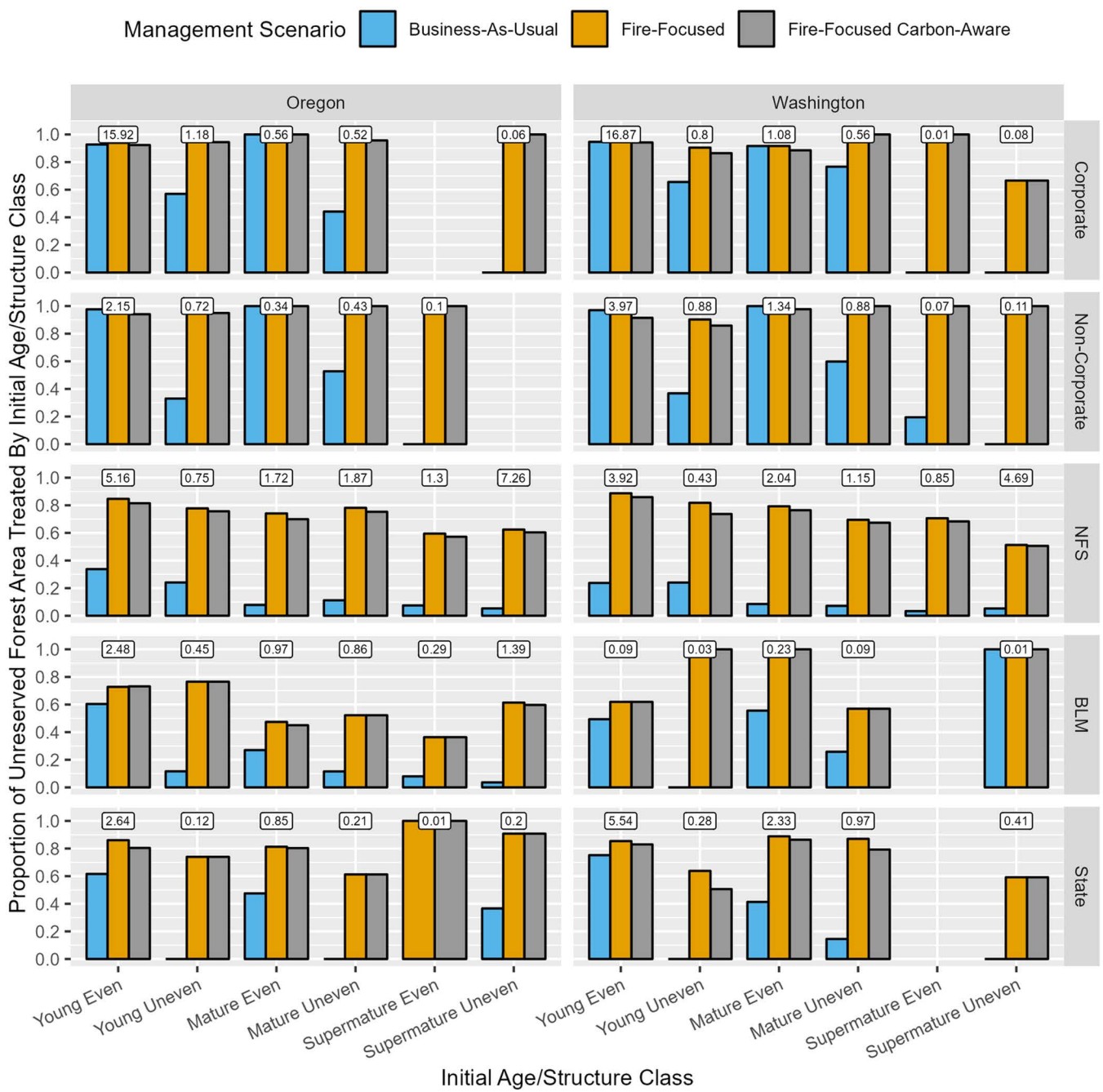

**Fig 3. Proportion of unreserved forest area treated by state (4.05 and 4.45 million hectares in Oregon and Washington, respectively), initial age/structure class, ownership, and forest management scenario.** Boxed values posted above bar cluster strata represent the percentage of total unreserved forest area in our study area associated with each stratum.

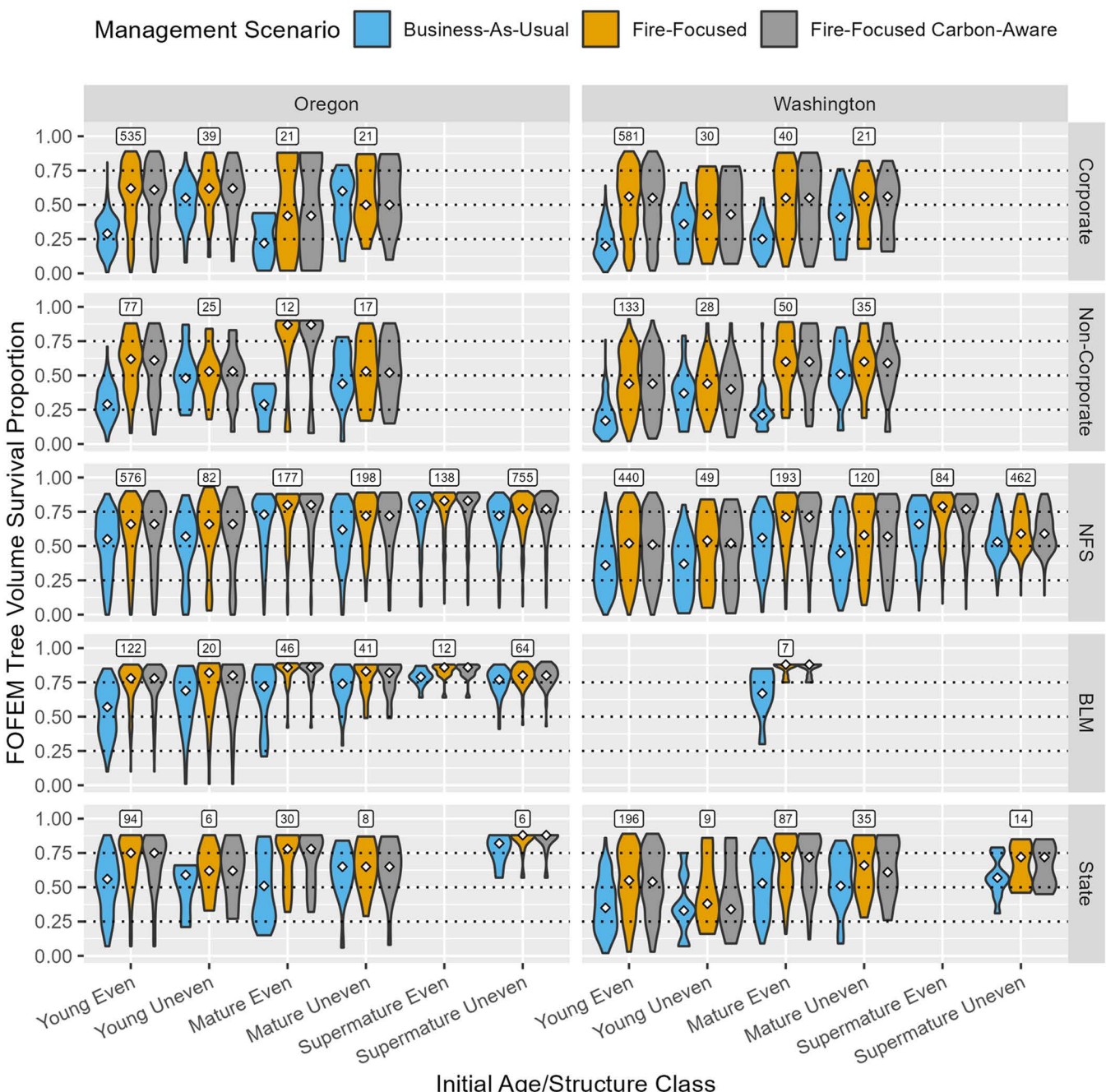

**Fig 4. Stand-scale fire resistance outcomes, represented by FOFEM-predicted tree volume survival proportion temporally weighted over the 40-year simulation, summarized via violin plots by State, initial age/structure class, ownership, and management scenario.** White diamonds indicate the area-weighted median value across stands in each category. Boxed values posted above violin clusters show the sample size (number of stands) associated with each stratum; stratum with 5 or fewer sample stands are not included in this chart. Table 2 describes the response variable and its calculation.

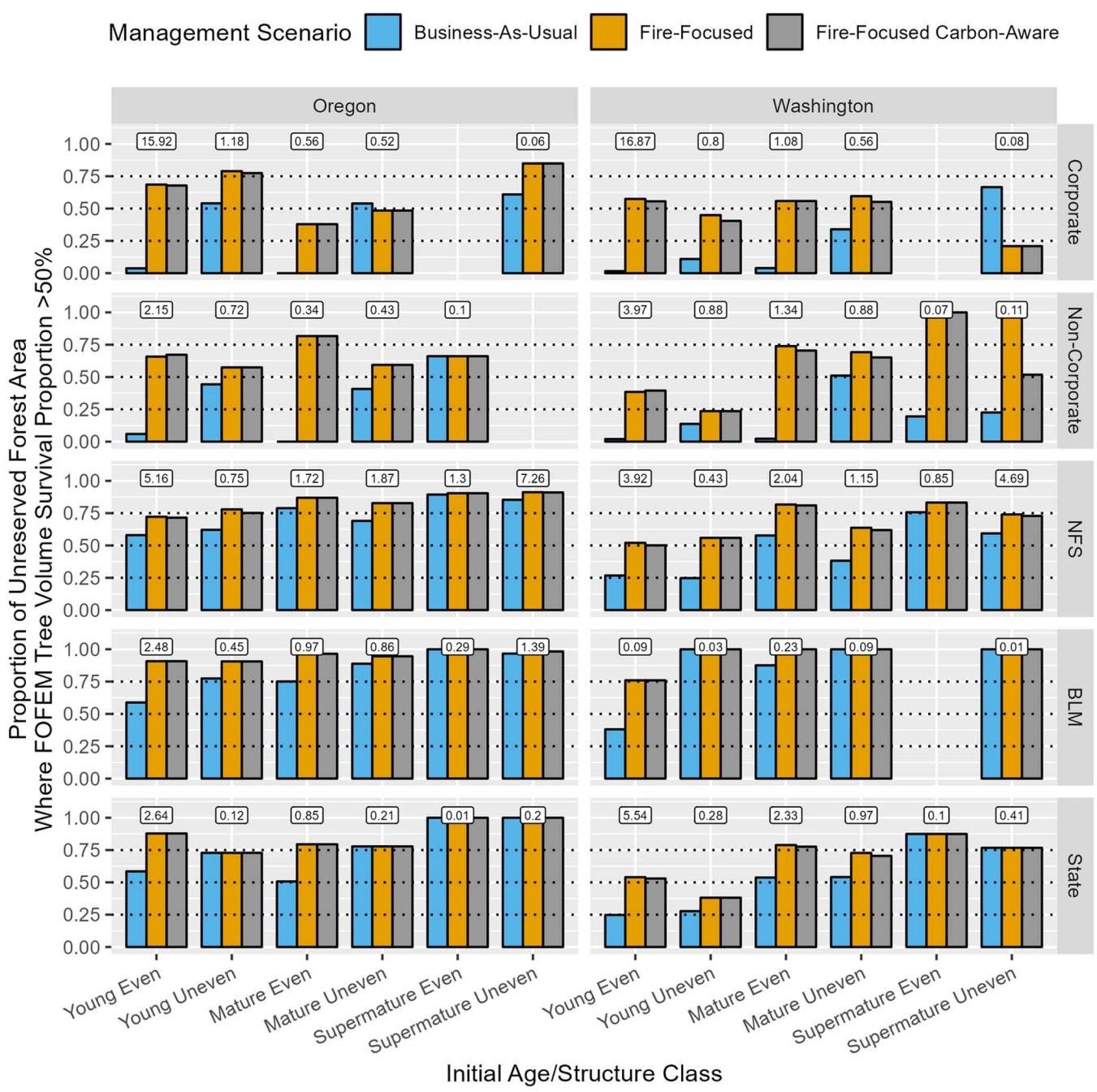

**Fig 5. Landscape-scale fire resistance outcomes, represented by the proportion of unreserved forest area where FOFEM-predicted tree volume survival proportion, weighted over the 40-year simulation, exceeds 50%, summarized via bar graphs by state, initial age/structure class, ownership, and management scenario.** Boxed values posted above bar cluster strata represent the percentage of total unreserved forest area in our study area associated with each stratum. Table 2 describes the response variable and its calculation.

FF and FFCA greatly increased the proportion of fire-resistant forest area on private lands relative to BAU (largely concentrated in young followed by mature even-aged forests; ~ 40%–80% increase in area), with more modest improvements observed in young followed by mature even-aged forests across public ownerships (~ 10%–40% increase in area).

FFE-TSVP exhibited outcomes similar to FOFEM-TSVP at stand and landscape scales, with some variability observed by ownership, age/structure class, and management scenario; FOFEM-TSVP typically predicted lower tree survival, likely owing to an assumed flame length greater than what the severe fire weather parameters produced under FFE-TSVP (S1 and S2 Figs). Relative to the FF management scenario, FFCA management exhibited similar or slightly lower fire resistance outcomes (0%–10% decrease in median value) as predicted by FFE-TSVP, likely a function of FFE's greater sensitivity to surface fuel loading dynamics. Fig 6 depicts plot level data in map form to show variation across the study area in fire resistance outcomes under BAU and the improvements in fire resistance that might flow from the FF management scenario.

## Carbon

At stand and landscape scales, over the 40-year simulation, less sequestration of carbon in-forest and in harvested wood products combined was observed under FF and FFCA management scenarios relative to BAU, and in some cases negative net sequestration was observed (Figs 7 and 8). This pattern held across initial age/structure classes and ownerships, except for young, even-aged stands in non-corporate ownership in Washington. In nearly all cases, greater sequestration was observed under FFCA vs. FF by removing from consideration all prescriptions that included pile burning or prescribed fire, thereby preventing carbon emissions from intentional burning practices. Reductions in net carbon sequestration when shifting from BAU to FF or FFCA were substantially more modest on private (~ 5%–20% decrease) than on public lands (~ 30%–300% decrease) at a landscape scale. The greatest rates of carbon sequestration were found in young, even-aged stands growing on productive sites, the majority of which are in Corporate private ownership.

Considering just the aboveground carbon stock change in live and dead trees between year 1 and year 41 of our simulations, FF and FFCA greatly reduced in-forest carbon on public lands (~ 50%–300% decrease) owing to sizeable reductions in what tended to be initially high stocking combined with re-treatments that maintained much lower stand densities, and thus tree stocking, over time. However, these management scenarios greatly increased forest carbon stocks in both young and mature forests in private ownership (~ 30%–120% increase), by shifting large proportions of the forest from short-rotation regeneration harvests to periodic selection harvests (S3 and S4 Figs).

## Economic

At the stand-scale and considering only the forest area receiving active management during the simulation, net present value (NPV) decreased under FF and FFCA management scenarios relative to BAU on private forests (particularly among mature even-aged stands), increased on NFS forests, and exhibited mixed outcomes on State and BLM forests (Fig 9). NPV was very high (~ 2,000–4,000 median 2,022 USD/ha/year) under BAU in many mature even-aged stands on State and BLM lands, which is attributed to clearcutting of stands early in the simulation with very high merchantable volumes and tree compositions dominated by large diameter Douglas-fir and western redcedar trees (i.e., the most valuable logs in our study area; S2 Table). While some forest area would generate negative NPV (i.e., value of a stream of future management costs, discounted to the present, exceeds the value of revenue from future sales of wood, also discounted), particularly under FF and FFCA management scenarios, for most forest area there are resistance-effective management options that generate a positive return. At the landscape-scale (Fig 10), NPV under FF and FFCA was primarily reduced (~ 20%–60% decrease) or unchanged relative to BAU in most owner and age/structure classes, except on national forests, where NPV increased by up to an order of magnitude (~ 120%–1000% increase), owing to the much larger proportion of forest area eligible for active management under FF and FFCA management scenarios (Fig 3).

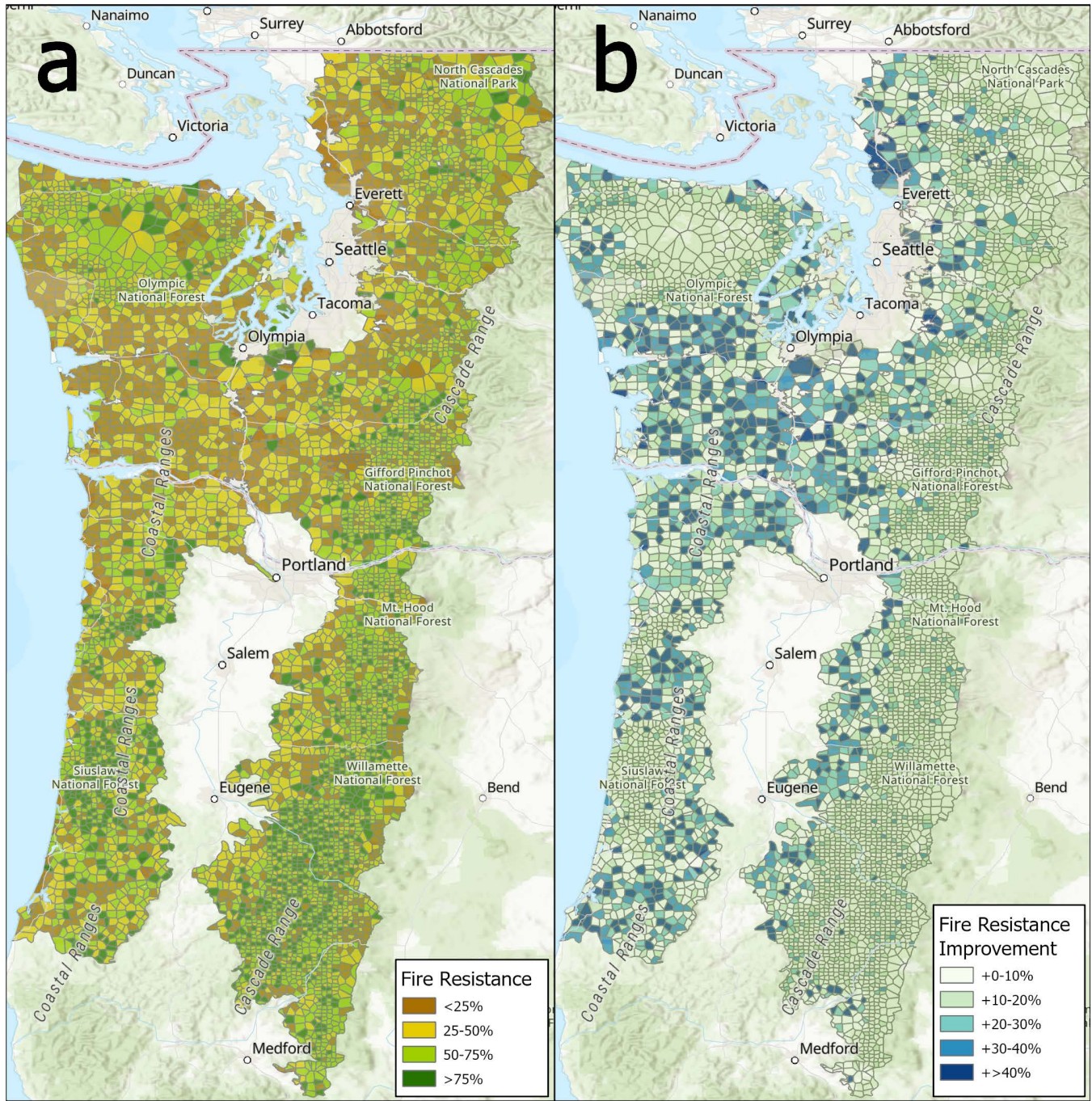

**Fig 6. Spatial representation of (a) First-Order-Fire-Effects-Model (FOFEM) predicted tree volume survival proportion expected to survive a wildfire encounter with 1.8-2.4m surface flame lengths (Fire Resistance, classified as percent survival) under the BAU management scenario and (b) the relative improvement in that percent survival that results from implementing the Fire-Focused management scenario across west-side forests in Oregon and Washington.** Thiessen polygons were formed around public (fuzzed) coordinates of FIA plots to show pattern in modeled fire resistance outcomes. Polygons are smaller where the FIA sample is intensified (on unreserved national forest land). Table 2 describes the response variable and its calculation.

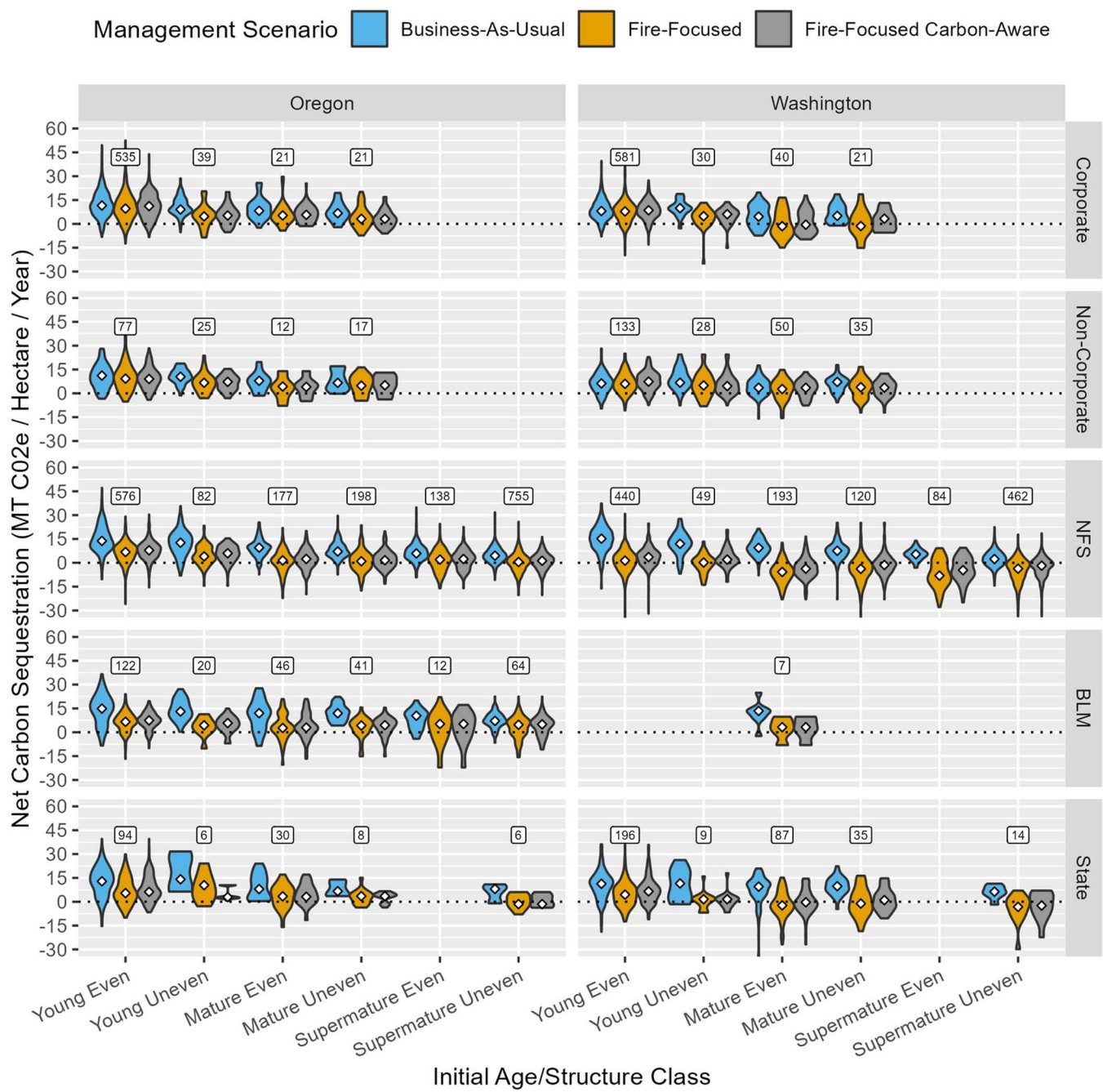

**Fig 7. Stand-scale, annualized net carbon sequestration (in live and dead wood and products minus fire emissions), summarized via violin plots by state, initial age/structure class, ownership, and management scenario.** White diamonds indicate the area-weighted median value across stands in each category. Boxed values above bar clusters indicate the sample size (number of stands) associated with each stratum; stratum with 5 or fewer stands are omitted. Table 2 describes the response variable and its calculation.

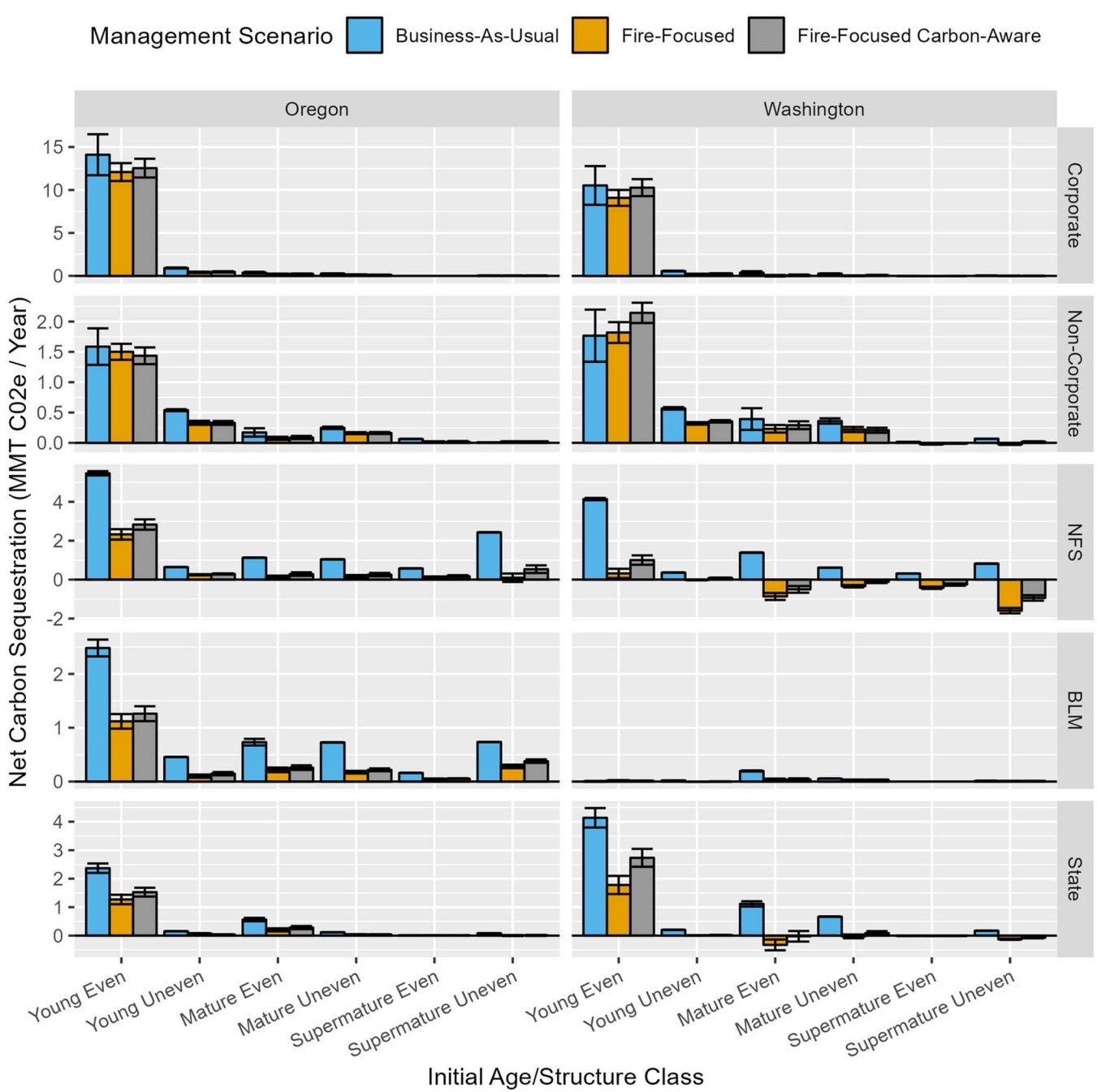

**Fig 8. Landscape-scale, annualized net (total) carbon sequestration (in live and dead wood and products minus fire emissions), summarized via bar graphs by state, initial age/structure class, ownership, and management scenario.** Bar values count 60% of harvested merchantable wood as sequestered by virtue of storage in long-lived (>100 yrs) wood products or via substitution benefits; lower and upper error bars reflect alternative assumptions about sequestration value of harvested merchantable wood (40% and 80%, respectively). Note that Y-axis scales differ by ownership. Table 2 describes the response variable and its calculation.

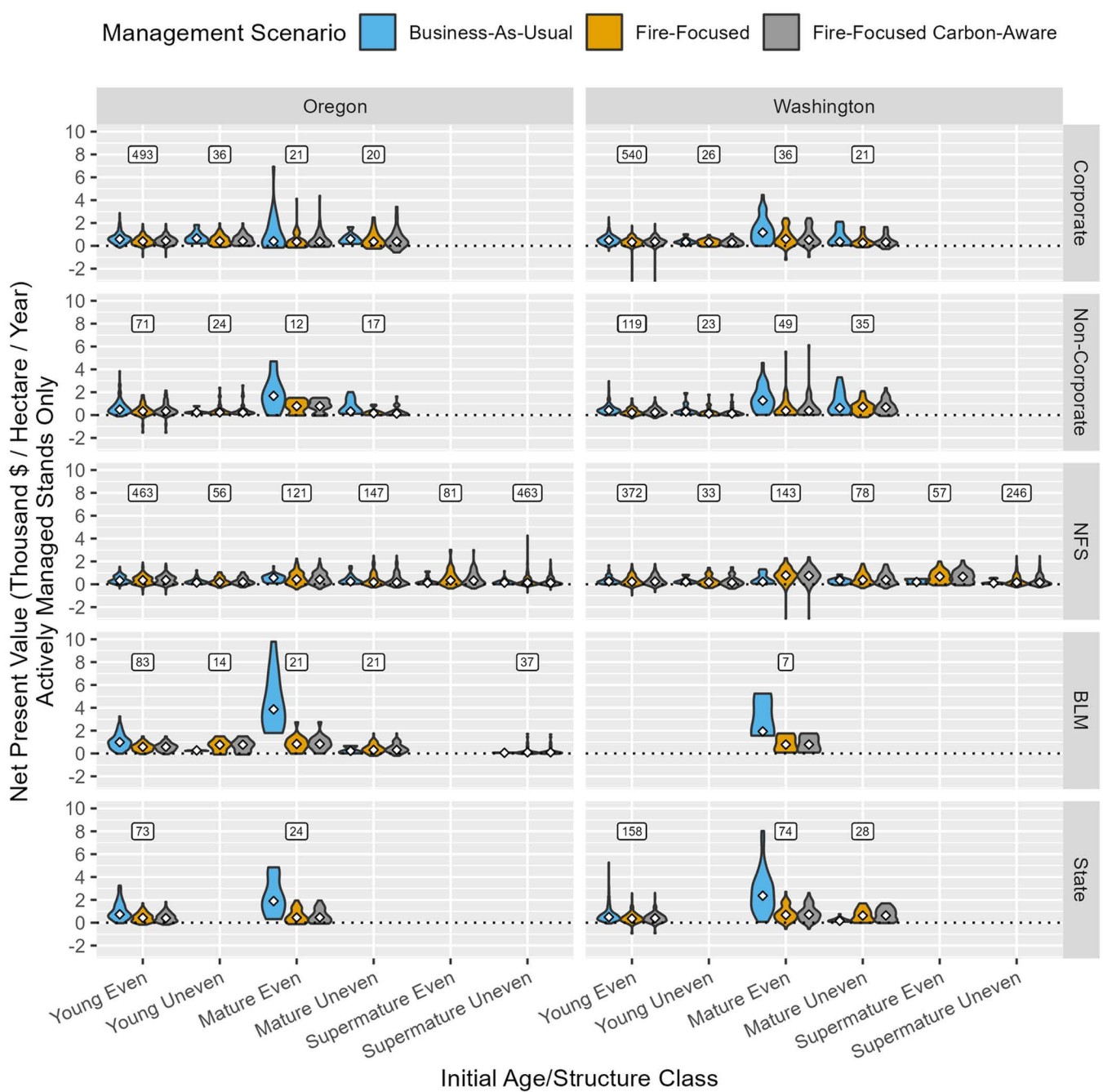

**Fig 9. Stand-scale net present value (calculated with a 4% discount rate) over the 40-year simulation period associated with forests receiving active management (at least one harvest entry), summarized as violin plots by state, initial age/structure class, ownership, and management scenario.** White diamonds show median values. Boxed values above violin clusters reference sample size (number of actively managed stands across management scenarios) associated with each stratum. Stratum with 5 or fewer stands are omitted. Net present value calculation is described in Table 2.

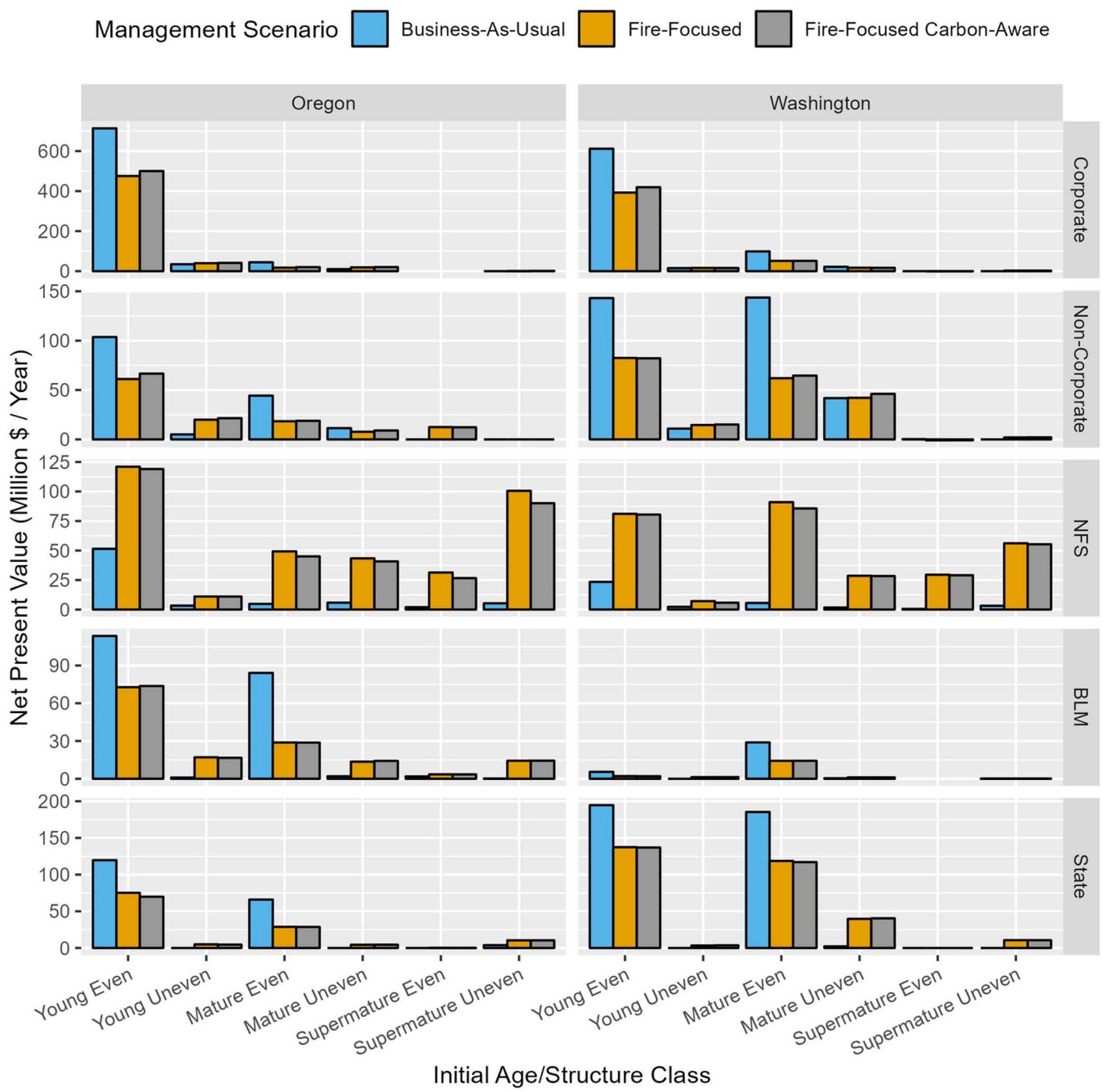

**Fig 10. Landscape-scale net present value (calculated with 4% discount rate) associated with active forest management, summarized as bar graphs by state, initial age/structure, ownership and management scenario.** Note that Y-axis scales differ by ownership. Net present value calculation is described in Table 2.

At the stand-scale, merchantable timber harvest volume generally decreased (~10%–100%) under FF and FFCA management scenarios relative to BAU (and especially among mature even-aged stands; ~100%–300% decrease), except on NFS lands, where it was similar or slightly higher; patterns in non-merchantable harvest volume varied more

widely by ownership and age/structure class (S5 and S7 Figs). At the landscape-scale, merchantable harvest volume was generally lower (~ 10%–30% decrease) under FF and FFCA relative to BAU, except on NFS lands, where it increased by at least an order of magnitude, from a very low base (S6 Fig). Non-merchantable harvest volume was lower or similar across ownership and age/structure class, except on NFS lands, where it, too, increased by at least an order of magnitude (S8 Fig).

## Tradeoffs

To support broadscale evaluation of stand-scale tradeoffs between potential improvements in fire resistance and associated carbon and economic outcomes under scenarios that prioritize fire resistance, we summarized area-weighted median and interquartile range across outcome metrics by initial age/structure class and management scenario (Table 3). With some exceptions, we found outcome metrics from FF and FFCA management scenarios to be statistically indistinguishable from one another but statistically different from BAU. Median improvement in fire resistance, as indicated

**Table 3. Stand-scale area-weighted medians (interquartile range) of fire resistance, carbon, wood harvest and financial outcomes, by initial age/structure class and management scenario, associated with unreserved forestland.**

| Initial Age/ Structure Class | Manage-ment Scenario | Area Actively Managed | Stand Scale Area-Weighted Median (Interquartile Range) Outcomes | | | | | |
|---|---|---|---|---|---|---|---|---|
| | | | FOFEM Tree Volume Survival Proportion | Carbon Sequestration (MT C02e/ha/ yr) | Change in Aboveground Carbon Storage (MT C02e/ha/yr) | Net Present Value* ($/ha/yr) | Merch Harvest Volume* (m3/ha/yr) | Non-Merch Harvest Volume* (m3/ha/yr) |
| **Young Even** | BAU | 79% | 0.29[a](0.18, 0.49) | 10.8[a](6, 17) | 5.2[a](1, 15) | 537[a](250, 798) | 13.5[a](8, 18) | 4.9[a](2, 6) |
| *2,758 Stands* | FF | 92% | 0.60[b](0.38, 0.75) | 7.3[b](1, 11) | 5.1[b](0, 9) | 352[b](100, 608) | 9.2[b](5, 13) | 4.2[b](3, 6) |
| *4,218,688 Ha* | FFCA | 89% | 0.60[b](0.36, 0.74) | 8.2[c](3, 12) | 5.6[b](0, 10) | 385[b](119, 626) | 9.9[c](6, 14) | 4.5[a](3, 7) |
| **Young Uneven** | BAU | 37% | 0.44[a](0.26, 0.61) | 10.2[a](7, 15) | 9.2[a](5, 15) | 319[a](196, 663) | 8.8[a](5, 13) | 4.0[a](2, 5) |
| *289 Stands* | FF | 87% | 0.57[b](0.35, 0.71) | 4.1[b](0, 8) | 2.8[b](−2, 6) | 262[b](59, 613) | 6.6[b](3, 12) | 4.2[a](2, 6) |
| *404,631 Ha* | FFCA | 84% | 0.56[b](0.31, 0.71) | 4.8[b](1, 9) | 2.9[b](−2, 6) | 264[b](64, 644) | 6.9[b](4, 12) | 4.4[a](3, 7) |
| **Mature Even** | BAU | 46% | 0.44[a](0.31, 0.73) | 8.3[a](4, 12) | 7.4[a](2, 12) | 1,536[a](614, 2,642) | 16[a](8, 25) | 7.0[a](3, 11) |
| *663 Stands* | FF | 84% | 0.74[b](0.56, 0.85) | −0.1[b](−6, 4) | −2.3[b](−10, 3) | 594[b](178, 1,056) | 9.3[b](5, 14) | 3.9[b](2, 6) |
| *824,382 Ha* | FFCA | 81% | 0.74[b](0.54, 0.85) | 1.0[c](−4, 5) | −1.8[b](−9, 4) | 595[b](183, 1,074) | 9.4[b](5, 14) | 4.0[b](2, 6) |
| **Mature Uneven** | BAU | 26% | 0.57[a](0.39, 0.72) | 7.9[a](4, 11) | 7.6[a](4, 11) | 383[a](182, 882) | 8.3[a](5, 12) | 4.2[a](1, 6) |
| *499 Stands* | FF | 81% | 0.66[b](0.49, 0.82) | 1.5[b](−4, 5) | −0.3[b](−7, 5) | 321[b](64, 784) | 7.0[b](3, 12) | 3.3[a](2, 6) |
| *541,202 Ha* | FFCA | 79% | 0.66[b](0.48, 0.81) | 2.2[c](−3, 5) | −0.1[b](−6, 4) | 351[ab](76, 811) | 7.0[ab](3, 12) | 3.5[a](2, 6) |
| **Superma-ture Even** | BAU | 6% | 0.74[a](0.59, 0.83) | 5.8[a](3, 9) | 6[a](3, 9) | 211[a](118, 447) | 5.6[a](4, 8) | 1.0[a](0, 2) |
| *245 Stands* | FF | 61% | 0.80[b](0.68, 0.86) | −0.7[b](−7, 5) | −2.1[b](−11, 4) | 445[b](84, 918) | 6.5[a](2, 12) | 2.2[b](1, 4) |
| *196,206 Ha* | FFCA | 59% | 0.80[b](0.67, 0.86) | 0.7[b](−4, 5) | −0.9[b](−8, 5) | 391[b](64, 869) | 6.4[a](2, 12) | 2.2[b](1, 4) |
| **Superma-ture Uneven** | BAU | 5% | 0.66[a](0.51, 0.77) | 4.2[a](1, 7) | 4.6[a](1, 8) | 129[a](48, 271) | 3.7[a](2, 6) | 1.1[a](0, 2) |
| *1,313 Stands* | FF | 59% | 0.72[b](0.58, 0.81) | −0.5[b](−5, 3) | −0.7[b](−6, 3) | 150[a](12, 506) | 3.9[a](1, 8) | 1.7[b](1, 3) |
| *1,023,565 Ha* | FFCA | 58% | 0.72[b](0.57, 0.81) | 0.2[c](−4, 4) | −0.2[b](−5, 3) | 148[a](13, 505) | 4.0[a](2, 8) | 1.8[b](1, 3) |

*Stands receiving active management only. Outcomes were computed over the 40-year simulation as weighted means, sums, or change over time - see Table 2 for more detail on calculation methods. Superscript letters indicate statistical differences (p < 0.05) in each Outcome variable among Management Scenarios, within Initial Age/Structure Class, based on pairwise Wilcoxon Rank Sum tests. Acronyms: BAU = Business-As-Usual; FF = Fire-Focused; FFCA = Fire-Focused Carbon-Aware; FOFEM = First Order Fire Effects Model; MT = metric tons.

by FOFEM-TVSP, was greatest in young and mature even-aged stands (~ 30% increase in median value). This benefit comes at the cost of reductions in merchantable and non-merchantable harvest volume and in the net present value (~ 18%–60% decrease in median value) effects linked to those reductions, and the very considerable reductions in carbon sequestration expected during the 40-year simulation period (~ 30%–800% decrease in median value). Fire resistance was enhanced to a lesser degree in young and mature uneven-aged forests (~ 9%–13% increase in median value) and accompanied by modest reductions in net present value, harvest volume, and carbon sequestration. Fire resistance improvements (~ 6% increase in median value) in supermature stands were even more limited and accompanied by a large reduction in carbon sequestration (~ 500% decrease in median value) and an increase in harvest volume and net present value. Median values of aboveground carbon storage in live and dead wood was reduced by up to an order of magnitude under FF and FFCA versus BAU, except in young even-aged stands, where similar median values were observed yet statistical differences were detected.

## Discussion

### Fire resistance improvement varies by initial forest structure

We crafted fire-aware treatments, available in the fire-focused (FF) and fire-focused carbon-aware (FFCA) management scenarios, as a departure from business-as-usual (BAU) silviculture that would elevate fire resistance, by disrupting vertical and horizontal crown fuel connectivity and reduce surface fuel loadings by removing or burning residues from harvest operations. While we simulated both even- and uneven-age fire-aware management, uneven-age treatments (i.e., selection harvest) tended to maximize fire resistance by 1) facilitating movement of a greater proportion of stand stocking and tree volume into larger, more fire-resistant trees and 2) largely avoiding the young and short statured stage of stand development that strongly facilitates crown fire initiation. Prescribed fire that reduced surface and ladder fuels post-thinning typically achieved the greatest fire resistance, an outcome with strong empirical support across drier forests elsewhere [30,45,46]. Resistance, however, was only slightly reduced if instead of burning and without regard to economic optimality, harvest residues were removed and utilized, generating less carbon emissions and smoke.

Although our results suggest fire resistance can be enhanced almost everywhere, if deemed a priority, the magnitude of enhancement varies substantially by initial stand age/structure class nested within ownership, and to some degree, State. Only incremental changes in resistance were seen in stands that are already relatively fire resistant (e.g., where at least half of live tree volume is predicted to survive wildfire burning under severe but less-than-extreme fire weather), while the changes under the FF and FFCA management scenario were substantial where resistance is currently low (e.g., where less than a quarter of live tree volume survives). Forests for which the BLM is responsible were the most fire resistant, followed by NFS (but see greater resistance in Oregon vs. Washington). State level differences may in part be explained by shifts in climate conditions along a latitudinal gradient, that favor higher tree densities and a higher proportion of species with lower fire resistance (e.g., western hemlock) in Washington, relative to Oregon [47]. Private forests were the least fire resistant, with almost no resistance in the young, even-aged, planted forests that dominate this ownership, which is consistent with findings from field and remote sensing studies that suggest fire effects are more severe in these forests than in older and more structurally diverse stands under less-than-extreme fire weather [13,48,49]. Under the FF management scenario, this age/structure class would see the greatest improvement in fire resistance via 1) reduced stand density (and thus horizontal fuel continuity) and surface fuels (and thus flame lengths), and 2) an increasing share of tree biomass in older and larger size classes which, particularly for Douglas-fir, the most common species in the region, equates to greater vertical separation between canopy base and surface fuels and thicker bark that protects from cambial injury [35].

The variation in fire resistance outcomes we observed across age/structure classes tracks a growing body of knowledge regarding the influence of forest structure on historical fire regimes and burn severity patterns. While large, infrequent, stand-replacing "megafires" that burn under extreme fire weather conditions and irrespective of forest structure are an often-searingly memorable component of the regional fire regime [12], two other patterns have been observed

under less-than-extreme fire weather. One is persistent, high-severity, short-interval reburning in young forests that were initiated by either stand-replacing fire or intensive, even-aged management (e.g., Tillamook and Yacolt fires [6,14]). The other is mixed-severity fires of varying (but generally smaller) size across uneven-aged mature to old-growth forests [3–4]. This second fire pattern historically reduced forest density and fuel loadings, while maintaining structural diversity and landscape-scale fire resistance under less-than-extreme fire weather. Allowing wildfires to burn under mild-moderate fire weather conditions across these more complex stands could reintroduce the positive benefits of fire at a landscape scale [32], which has been largely suppressed out of existence since the mid-20th century, except in designated wilderness and sometimes forests immediately adjacent to wilderness [6]. Further, encouraging the transition of a larger proportion of private forests into these older and more complex age/structure classes, while implementing a spatial arrangement that intermingles these kinds of stands with younger forests vulnerable to stand-replacement, could 1) reduce the size and extent of future stand-replacing fires, 2) expand ecosystem and habitat benefits more broadly, and 3) reduce risk of wild-fire spread into or from adjacent human population centers (i.e., the wildland urban interface intersecting the Willamette Valley and Puget Lowlands; Fig 6).

## Enhanced fire resistance implies trade-offs in economic and carbon benefits

The essence of enhanced fire resistance is reduced tree density and surface fuel loadings, which imply 1) reduced in-forest carbon stocks under a fuels treatment program and 2) potentially lower carbon sequestration, given that reduced site occupancy may leave some resources (light, water, nutrients) unutilized, such that cumulative stand growth is not maximized. Assuming a large proportion of harvested trees are processed into long-lived wood products, traditional, fire-unaware, intensive even-aged management not only maximizes economic objectives – it maximizes carbon seques-tration by maximizing growth, albeit capping in-forest carbon storage to a typically short rotation age (~ 45 years) that does not maximize mean annual increment. Our results, among other studies at a global scale [50], suggest that this management paradigm, however, results in far greater risk of stand-replacing fire, and thus forests managed under this paradigm only maximize economic and carbon sequestration benefits while they remain effectively fire-suppressed. While regional fire suppression efforts over the mid to late 20th century have been largely successful at maintaining this par-adigm, that may not continue. Increasing wildfire activity during the early 21st century in the western United States was accompanied by substantially increased investment in firefighting resources, yet the fire protection system still falls far short of the historical policy goal of 100% suppression [51]. Fire activity has been increasing in Pacific Northwest for-ests during this period [8] and further increases are expected by the end of the century [9–10]. As these fuel-rich forests become increasingly dry nearly every year, and thus primed for wildfire, fire suppression may no longer be the only viable strategy to mitigate the negative social, ecological, and environmental outcomes of large stand-replacing fires.

Our results also suggest that FF and FFCA management of young even-aged stands on Corporate and Non-Corporate private forests may garner the greatest benefits without significantly compromising economic and carbon goals, relative to other age/structure classes and ownerships. Under these management scenarios, carbon sequestration decreases slightly (except on Non-Corporate forests in Washington, where it increased) at a landscape scale, and net present value and harvest volume decrease moderately, while in-forest carbon storage increases moderately to considerably, accom-panied by considerable increases in fire resistance (i.e., expansion of the forested area where more than 50% of tree volume is predicted to survive a severe weather wildfire). Some losses in net present value could be mitigated through more selective and strategic utilization of non-merchantable harvest residues; we assumed those residues would always be utilized regardless of haul costs. On other ownerships, achieving relatively modest increases in fire resistance reduces harvest volume and net present value, except on national forests, where both increase dramatically owing to the much larger share of forest area assumed available for active management under the management scenario. Carbon storage and sequestration are dramatically reduced for all age/structure classes over the 40-year simulation, because lower stock-ing, and capacity to absorb carbon, is maintained irrespective of whether a stand burns over those four decades.

While treatment enhances survival of trees in stands that do burn, most of the carbon in fire killed trees (of which there are many more in an untreated stand) will have not yet emitted to the atmosphere by the end of the simulation, which is only 10 or 30 years after the modeled burn year. Lacking the capacity to accurately project stand outcomes to the point where all carbon in fire-killed trees has emitted poses challenges for attaining definitive conclusions about differences in carbon dynamics between managed and unmanaged realizations of a stand that burns. Assuming FF and FFCA management scenarios are effective at mitigating stand-replacing fire in the medium-to-long-term and at a landscape scale, carbon storage and sequestration outcomes could reverse (i.e., increase) over longer temporal periods (i.e., next 80 + years) relative to BAU [52].

It is worth noting that the assumption of active management on most national forest land is contrary to agency practice over the past few decades, during which timber harvest has been relatively rare at a landscape scale. The clear differences in fire resistance outcomes and tradeoffs by ownership suggest that while broadscale adoption of fire-aware management across young even-aged forests on Corporate and Non-Corporate private lands may meaningfully reduce landscape scale fire hazard, with only modest tradeoffs in carbon and economic benefits, the same cannot be said about public ownerships. Rather, managers of public lands will need to carefully and strategically evaluate where thinning and fuel treatments can meaningfully reduce fire risk and protect critical assets (social and/or biological [53]), given the potentially substantial carbon and/or economic tradeoffs of doing so. Where treatment benefits justify those tradeoffs on public lands, and especially on national forests, harvest residues from these treatments could provide a sizable feedstock for the expansion of regional biomass and biochar facilities, which could reduce carbon emissions associated with such treatments.

## Limitations and uncertainty

Our findings depend on 1) the Forest Vegetation Simulator's (FVS) representation of tree growth, mortality, fire effects, and fire-killed wood decay over 40-years; our assumptions about 2) the amount of harvested wood allocated to wood products with a long-term carbon storage benefit; and 3) the extent to which energy generated from utilized forest residues replaces fossil fuel-based power. Further, the management scenarios we crafted, simulated, and report on involve a suite of decision points, generalizations of management behavior, and in the case of the FF and FFCA management scenarios, better capture what is possible than what is probable. Thus, our results should be carefully interpreted within the context of those decision points and management scenarios.

While FVS is built upon empirical data and relationships, only parts of FVS have been formally validated by the developers and/or by independent researchers. FVS is known to overpredict growth and underpredict mortality without parameterization [37,54,55]. We used best practices to mitigate this behavior (e.g., calculating and supplying MaxSDI values, potential vegetation type codes, using tree diameter and growth remeasurement data for growth model calibration, and systematically accounting for natural tree regeneration dynamics). We did not directly account for the impact of climate warming on tree growth and mortality in FVS over time (i.e., assumed static climate), but did account for the impact of climate warming and decreasing fire suppression effectiveness on increasing wildfire activity, via inclusion of a wildfire prevalence scenario assuming 5% area burned per decade – a rate greater than observed in recent history [8]. Further, we chose to limit our simulation timeline to 40-years, to reduce multiple compounding sources of internal (model-based) and external (climate warming, changes in disturbance regimes and forest management) uncertainty in our results. If those sources of uncertainty could be sufficiently accounted for, however, a longer simulation timeline (e.g., 80–100 years) could better capture the long-term carbon trajectories of forests. Our 40-year simulation period does not fully capture carbon emissions from fire-killed dead wood decay, and the effects of stand-replacing fire realized under a BAU management scenario could limit and/or slow long-term carbon storage recovery and accumulation (i.e., natural tree regeneration) when post-fire live seed source availability is limited and/or stands are affected by subsequent short-interval fire(s) and/or other disturbances [56–57].

The specific parameters and assumptions associated with harvested wood product life cycle assessment (LCA) analyses can significantly impact carbon accounting outcomes. We did not conduct or use an existing LCA for our study region and instead used simplistic assumptions about harvested wood product carbon storage, which could grossly understate stable carbon storage in harvested wood, in a future where carbon management is a higher priority than it is today. If technology and support for utilization of non-merchantable residues and small diameter logs into engineered wood products continue to advance, a larger proportion of that material can be stored in long-lived wood products, reducing carbon emissions implied by harvest. Finally, we didn't formally account for any substitution benefits (from using wood instead of other building materials with higher embodied carbon), given it is difficult to know what products the harvested wood will become today, let alone over the next few decades, so some avoided carbon emissions that result from how harvested wood products are used may be understated.

## Conclusions

As wildfire activity continues to increase across mesic westside Pacific Northwest forests, management tools other than fire suppression alone are needed to strategically mitigate the risk and extent of large stand-replacing wildfires under less-than-extreme fire weather conditions. Our study suggests that fire-focused (FF) and fire-focused carbon-aware (FFCA) management and fire-aware silvicultural prescriptions can meaningfully enhance regional fire resistance over the next 40-years at stand and landscape scales, relative to business-as-usual practices, most compellingly where current fire resistance is currently poor: young even-aged forests and especially those concentrated on privately-owned lands. Alternative FF and FFCA management in these forests, which make up about half of the westside landscape, can increase the areal proportion of these stands in a fire-resistant state by ~60% in Oregon and ~40% in Washington by mid-21st century. Reduced carbon sequestration and economic returns realized through fire-focused management, relative to business-as-usual, are also marginal to modest for this forest demographic, relative to other intersections of age/structure class, ownership, and State. Across public ownerships, FF and FFCA management can still play an important role in strategically mitigating fire risk near critical social, ecological, or environmental resources, but doing so can modestly to dramatically reduce medium-term aboveground carbon storage and sequestration, and in some cases economic returns. If FF and FFCA management can substantially reduce the extent of stand-replacing fire at a landscape scale over the medium- to long-term, however, long-term carbon storage and sequestration outcomes may outweigh potential medium-term losses. Given the current lack of empirical studies focused on quantifying fuel treatment effectiveness across mesic westside Pacific Northwest forests, these findings may help land managers strategically mitigate fire risk while weighing the carbon and economic tradeoffs of doing so.

## Supporting information

**S1 Appendix. Additional methodological information associated with the Forest inventory data and Silvicultural treatments sections.**
(DOCX)

**S1 Fig. Stand-scale fire resistance outcomes, represented by FFE-predicted tree volume survival proportion temporally weighted over the 40-year simulation, summarized via violin plots by state, initial age/structure class, ownership, and management scenario.** White diamonds indicate the area-weighted median value across stands in each category. Boxed values posted above violin clusters show the sample size (number of stands) associated with each stratum; stratum with 5 or fewer sample stands are not included in this chart. Table 2 describes the response variable and its calculation.
(TIF)

**S2 Fig. Landscape-scale fire resistance outcomes, represented by the proportion of unreserved forest area where FFE-predicted tree volume survival to wildfire, weighted over the 40-year simulation, exceeds 50%, summarized via bar graphs by state, initial age/structure class, ownership, and management scenario.** Boxed values posted above bar cluster strata represent the percent of total unreserved forest area associated with each stratum. Table 2 describes the response variable and its calculation.
(TIF)

**S3 Fig. Stand-scale, annualized net change in aboveground carbon storage, excluding harvested wood storage, over the 40-year simulation, summarized via violin plots by state, initial age/structure class, ownership, and management scenario.** White diamonds indicate the area-weighted median value across stands in each category. Boxed values posted above violin clusters show the sample size (number of stands) associated with each stratum; stratum with 5 or fewer sample stands are not included in this chart. Table 2 describes the response variable and its calculation.
(TIF)

**S4 Fig. Landscape-scale, annualized net change in aboveground carbon storage, excluding harvested wood storage, summarized via bar graphs by state, initial age/structure class, ownership, and management scenario.** Note that Y-axis scales differ by ownership. Table 2 describes the response variable and its calculation.
(TIF)

**S5 Fig. Stand-scale merchantable harvest volume over the 40-year simulation period among forested stands that experienced active management (at least one harvest entry), summarized as violin plots by state, initial age/structure, ownership, and management scenario.** White diamonds indicate the area-weighted median value across stands in each category. Boxed values posted above violin clusters show the sample size (number of actively managed stands across management scenarios) associated with each stratum; stratum with 5 or fewer sample stands are not included in this chart. Table 2 describes the response variable and its calculation.
(TIF)

**S6 Fig. Landscape-scale, annualized merchantable harvest volume associated with active forest management, summarized via bar graphs by initial age/structure class, ownership, and management scenario.** Note that Y-axis scales differ by ownership. Table 2 describes the response variable and its calculation.
(TIF)

**S7 Fig. Stand-scale non-merchantable harvest volume over the 40-year simulation period among forested stands that experienced active management (at least one stand entry), summarized via violin plots by initial age/structure class, ownership, and management scenario.** White dots indicate the area-weighted median value across stands in each category. Boxed values posted above violin clusters show the sample size (number of actively managed stands across management scenarios) associated with each stratum; stratum with 5 or fewer sample stands are not included in this chart. Table 2 describes the response variable and its calculation.
(TIF)

**S8 Fig. Landscape-scale, annualized non-merchantable harvest volume associated with active forest management, summarized via bar graphs by state, initial age/structure class, ownership, and management scenario.** Note that Y-axis scales differ by ownership. Table 2 describes the response variable and its calculation.
(TIF)

**S1 Table. Additional harvest costs, by treatment type, used to parameterize the Processor module of BioSum.** Costs were sourced from regional foresters and fuel treatment experts.
(DOCX)

**S2 Table. Delivered timber values by tree diameter range and species group, averaged across subregions, used to parameterize the Processor module of BioSum.** Values were estimated by a forest economist (Mike Buffo, Mason, Bruce and Girard) using the MBGTools model.
(DOCX)

**S3 Table. Wildfire simulation weights used to calculate weighted average values at each simulation timestep, across simulation output metrics, to represent a percent area burned per decade fire prevalence scenario.** Fire effects were modeled in the Forest Vegetation Simulator (FVS) using the Fire and Fuels Extension (FFE).
(DOCX)

**S4 Table. Silvicultural treatment selection optimization logic by ownership and initial age/structure class associated with the business-as-usual (BAU) and fire-focused (FF) forest management scenarios.**
(DOCX)

## Acknowledgments

We thank G. Engbring, S. Charnley, and E. White for providing data associated with the regional large landowner surveys and interviews that we used to parameterize silvicultural treatments and management scenarios. We're grateful to M. Buffo, who provided regional timber value information and J. McGovern, who provided silvicultural fuel treatment cost estimates used to parameterize BioSum's economic model. R. Darbyshire, J. Sherlock, and M. Ritchie provided critical feedback that aided in crafting and refining the silvicultural treatments simulated in this study. D. Gatziolis contributed technical improvements to BioSum's calculation and estimation of travel times (from FIA plots to mill locations) and haul costs for this study. We acknowledge and are indebted to the dozens of Forest Service and contractor inventory field crew who, for over two decades, collected and quality-assured the data we relied on and to the programmers responsible for developing the BioSum modeling framework, including L. Potts, L. Bross and D. Lindstrom.

## Author contributions

**Conceptualization:** Jeremy S. Fried.

**Data curation:** Sebastian Busby.

**Formal analysis:** Sebastian Busby.

**Funding acquisition:** Jeremy S. Fried.

**Investigation:** Sebastian Busby.

**Methodology:** Sebastian Busby, Jeremy S. Fried.

**Project administration:** Jeremy S. Fried.

**Software:** Sebastian Busby.

**Supervision:** Jeremy S. Fried.

**Visualization:** Sebastian Busby.

**Writing – original draft:** Sebastian Busby.

**Writing – review & editing:** Jeremy S. Fried.

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
