## [Decision Letter · Decision Letter 0]

16 Jun 2025

Dear Dr. Busby,

Thank you for submitting your manuscript to PLOS ONE. After careful consideration, we feel that it has merit but does not fully meet PLOS ONE’s publication criteria as it currently stands. Therefore, we invite you to submit a revised version of the manuscript that addresses the points raised during the review process.

We look forward to receiving your revised manuscript.

Kind regards,

Julia A. Jones

Academic Editor

PLOS ONE

 [This research was supported by the U.S. Department of Agriculture (USDA) Forest Service PNW Research Station’s Westside Fire Research Initiative, which funded appointments to the Research Participation Program administered by the Oak Ridge Institute for Science and Education (ORISE) through an interagency agreement with the U.S. Department of Energy (DOE). ORISE is managed by ORAU under DOE contract number DE-SC0014664. The findings and conclusions in this paper are the responsibility of the authors and should not be construed to represent any official view, determination or policy of the USDA, DOE, ORAU/ORISE or any other U.S. Government entity.]. 

Additional Editor Comments:

Both reviewers recommend major revisions, with complementary review comments. Please ensure that you respond to every review comment in your revision and provide a rebuttal letter detailing each response.

Reviewers' comments:

Reviewer's Responses to Questions

**Comments to the Author**

1. Is the manuscript technically sound, and do the data support the conclusions?

Reviewer #1: Yes

Reviewer #2: Yes

2. Has the statistical analysis been performed appropriately and rigorously?

Reviewer #1: No

Reviewer #2: No

3. Have the authors made all data underlying the findings in their manuscript fully available?

Reviewer #1: Yes

Reviewer #2: Yes

4. Is the manuscript presented in an intelligible fashion and written in standard English?

Reviewer #1: Yes

Reviewer #2: Yes

Reviewer #1: This manuscript addresses a significant knowledge gap related to the impacts of fuel treatments or “fire aware” management in moist, westside forests of the PNW. As such, it has significant value to land management planning in the region. The authors modeling approach appears to be appropriate, although there is perhaps room for added analysis of results that are presented as stand-scale averages.

Throughout: this is a minor point, but when referring to categories of forest management or silvicultural treatments, the commonly-used terminology is “even-aged” and “uneven-aged” rather than “even-age” and uneven-age”. See for instance, Nyland, RD. 2016. Silviculture: concepts and applications. 3rd Ed. Waveland or Deal, R. 2017. Dictionary of Forestry. Society of American Foresters.

Introduction General: I suggest adding a paragraph (or even just a couple sentences somewhere) to highlight the types and reported effects of commonly-applied fuel reduction/fire resilience treatments that have broad support in other regions/ecosystems as a way to (1) provide context and support for the “fire-aware silvicultural treatments” you included in your simulations, and (2) highlight the discrepancy between common conceptions (misconceptions?) about the potential efficacy of fuel treatments in moist, westside forests of the PNW and a literature base that suggests some treatments (e.g., broadcast burning plus thinning) pretty consistently reduce fire severity across a wide array of biophysical environments outside of the most extreme fire weather conditions.

L75-94: It might be helpful to provide a figure (e.g., a pie chart) that shows the land ownership breakdowns described here. The information is provided in the text, but

L79-82: Is active forest management on NFS lands rare, or just constrained to a small percentage of the landbase (i.e, primarily in plantations that fall within NWFP matrix and AMA land use allocations), and a limited set of activities that have relatively broad social license (i.e., thinning, pile burning, occasion broadcast burning, and planting after wildfires)?

L93-94: It is misleading to imply that the term “regeneration harvest” equates to clearcutting. Clearcutting is the regeneration harvest method that is overwhelmingly used on industrial forestlands in the PNW, but there are many other regeneration harvest methods (e.g., seed tree cuts, variations on shelterwoods, retention harvests, and several variants on selection cutting methods). Consider something along the lines of “plantations of Douglas-fir… managed using clearcut harvests on approximately 35-60 year rotations” to avoid perpetuating the misconception that a regeneration harvest always means a clearcut.

L95-96: The sentiment of this statement is accurate (i.e., that fuel management techniques have not been widely utilized in moist, westside forests of the PNW), with one exception: slash burning is commonly practiced after clearcuts and sometimes practiced after thinning within this region. While this certainly doesn’t represent a comprehensive fuel management program, and may be done as much to facilitate planting as to reduce fuel loading, pile burning (and the broadcast burning of slash that was common in the region prior to the 1980s) is a common practice with a supporting research based that goes back to at least the 1950’s (e.g., Morris, M.G. 1958. Influence of slash burning on regeneration, other plant cover, and fire hazard in the Douglas-fir region. Research Paper No. 29. USDA Forest Service Forest and Range Experiment Station. Portland, OR). I suggest simply acknowledging that there has been relatively common use of slash removal as a site preparation technique following clearcut harvests in the region (perhaps referencing a couple of associated studies), while fuel treatments in existing stands have been rare.

L269-272: I suggest a minor wording revision to clarify that the silvicultural practices described were the examples of uneven-aged and even-aged silvicultural practices that you chose to simulate. For instance, “Treatments were associated with representative examples of either even- or uneven-age management styles, where the former was simulated using clearcutting (both with and without commercial thinning?) on varied rotations and the latter was simulated using repeated individual tree selection harvests, with trees thinned evenly across diameter classes in each entry, to maintain recruitment of new, younger growing stock over time.” Please note that my use of the terminology “individual tree selection harvests” instead of thinning would be appropriate if your repeated entries were associated with regeneration establishment objectives. If not, then “repeated commercial thinning” is fine, but isn’t actually representative of uneven-aged management. It would just be repeated thinning.

Methods and Connections to Results (General): Several of the primary results describe stand-level outcomes, while others describe landscape-level outcomes that are reflective of proportions of stands that meet some criteria, or (for NPV) presumably the cumulative NPV across all stands. This is fine, but I don’t see a clear rationale for why the results that represent stand-scale statistics (i.e., means of stand-level values) haven’t been compared with an appropriate statistical model. The authors should provide either some rationale for this choice, or adopt an appropriate statistical modelling framework to support inference from comparisons of stand-scale means. A GLM framework (or perhaps LMM with random effects for ecoregion or some other broad quantifier of spatial and biophysical proximity) seems perfectly acceptable here.

L655-657: The claim here that uneven-aged treatments produced less vertical canopy continuity is puzzling. In general, I would expect even-aged stand structures to have more uniform, single-strata canopies, while the multi-cohort structures that result from uneven-aged management, and the associated variability in tree diameters, heights, and crown positions should tend to produce more vertical connectivity. Additionally, I would expect the initiation of repeated thinning to delay crown recession, which would also tend to promote connectivity between younger understory to midstory tree cohorts and an older cohort of larger trees. At least from a diameter distribution standpoint, multiple studies suggest uneven-aged silviculture results in more varied tree sizes than even-aged management (e.g., Goodburn and Lorimer 1999 – Forest Ecology and Management, Janowiak et al. 2008 – Forest Science), and this is also implied by diameter distributions reported in several uneven-aged management studies in Douglas-fir forests of the PNW (Miller and Emmingham 2001, Ralston et al. 2004, Curtis 2010), one of which was co-authored by a co-author on this manuscript. In general, these more varied/less normally-distributed diameter distributions are associated with more varied tree heights and crown positions. Thus, I wouldn’t expect reduced vertical canopy continuity in stands managed with uneven-aged silvicultural approaches. Perhaps the authors could provide height distribution data to support this claim?

Alternatively, increased fire resistance across a population of stands managed using uneven-aged management, could be explained by other mechanisms. For instance, in uneven-aged systems all stands tend to have a significant proportion of their SDI (or volume since that was the metric used for characterizing fire resistance in this study) in larger, more fire resistant trees, while across a population of stands managed using even-aged techniques, a significant proportion of stands will always be younger stands with most of their SDI and volume in small-diameter trees with limited fire resistance. This provides what may, perhaps, be a more intuitive rationale for uneven-aged management to provide increased fire resistance at the landscape scale, and is consistent with later points in the discussion about the lack of fire resistance in young, even-aged plantations.

L728: Consider revising to read: “Under fire-focused management, carbon sequestration decreases slightly…

Reviewer #2: Please find a better-formatted version of the comments to authors attached.

Summary:

In this manuscript, Busby and Fried seek to characterize potential tradeoffs between fire resistance, carbon, and economic goals with regard to forest management strategies in the western Cascades, using a simulation modeling approach. The authors compare business-as-usual and fire-aware management approaches across a variety of land ownership and forest structure scenarios. The authors note potential considerations for future management based on these results.

I think that this study will be of interest to both researcher and manager readership of PLOS ONE, and is appropriate for this journal’s aims and scope. Overall I think this study is timely and well-designed, but would benefit from additional edits for clarity and conciseness before publication. I have outlined my recommendations below and look forward to seeing the next version of the manuscript.

Main comments (essential revisions):

1. At 26 pages, 15 figures, and 5 tables, the paper is quite long and in places feels unfocused. I can appreciate the challenge with concisely reporting findings for such a large, multidimensional model. That said, I suggest editing with an eye toward trimming the paper back to its main story. I have made some specific suggestions below, but consider limiting figures to one per results section (i.e., one carbon figure, one fire resistance figure, one economic figure), paneling or summarizing results as necessary. I would also consider actually adding a figure to the tradeoffs section (either in addition to or in place of the table). To me this is the “star” of the paper and it would be great to have a visual here that is digestible at a glance.

2. The First Order Fire Effects Model is a major part of this study. However, as far as I can tell the first mention of this model is in Table 3, with no information on how this model was used or parameterized in the methods. I feel that I cannot adequately evaluate this portion of the study without more information.

3. It is not clear to me whether each individual model scenario was run multiple times or just once. If multiple times, how much variability in outcomes is there per scenario? If just once, how did you account for stochastic effects in the model?

4. Although results are thoroughly described, the paper is lacking in statistical testing in general. It would be especially helpful to see some additional stats in the tradeoffs section of the results. Are outcomes significantly different by initial forest structure and/or management scenario?

Minor comments (desirable revisions):

1. The simulation period (40 years) does not include the full range of a single rotation period (35-60 years), which I feel is a crucial caveat to interpreting the results. While this is addressed in Lines 738-750 of the discussion, I would like to see a bit more discussion of the effects of simulation length on outcomes other than carbon alone (i.e., potential differences between short-term and long-term tradeoffs, especially in the context of repeat burns). Although this is touched on in the discussion, it may also be worth briefly noting in the methods (Line 295) why 40 years was chosen.

2. There are several sentences in the methods and results section that are difficult to parse – I have outlined a few in the following section. Consider breaking up clauses into sentences and reducing parentheses use where possible.

3. There are a great many acronyms in this paper. Although they make sense in cases where they are defined appropriately and used frequently (like BAU, FF, and FFCA), others (like LCA and FFE) are only used a handful of times and should be written out.

4. In general I would like to see more quantitative reporting in the results, especially if the number of figures is reduced. Statements such as “exceptionally low” (Line 490), “modest improvements” (Line 511), “lower tree survival” (Line 515), “more modest” (Line 535), etc. should be replaced with the corresponding value or statistic where feasible.

5. It was difficult for me to parse that the FF and FACC scenarios fall under the FA category presented in Table 1. I would recommend clarifying this in 397-399.

Specific text comments

• Lines 40-51: This paragraph lacks citations; I would recommend at least adding an appropriate citation for the sentence beginning with “Westside forests…” (lines 44-47).

• Lines 107-109: Not an edit – just noting that I really like this sentence.

• Lines 144-169: Some of this information is already stated in the introduction; I recommend condensing where possible.

• Lines 213-215: I am not sure what is meant by “used stand inventory conditions to select and enter an appropriate code”.

• Lines 220-247: This paragraph is very long – consider breaking up at line 231 (“Interviews with…”)

• Lines 251-254: I am not sure what is meant by this sentence.

• Line 300: Clarify why these wind speeds were selected.

• Lines 320-328: Consider moving these lines to the “Wildfire Activity and Prevalence” section.

• Lines 386: Why was Year 41 unevenly weighted?

• Line 475: Consider changing 18% unmanaged to 82% managed, for consistency and easier comparison with the preceding paragraphs.

• Line 550: Should this be FF instead of FA?

• Line 578 and elsewhere: Consider changing “stand level” and “landscape level” to “stand scale” and “landscape scale” for clarity.

• Lines 635-650: Lots of background information here before a reference to your findings – consider removing this paragraph altogether or moving to the introduction.

Figures & tables

• Figure 1 – the color schemes here are a bit confusing, as the colors for “State” and “Mature Even”, “Private” and “Young Uneven”, and “NFS” and “Supermature Even” are very similar. It is easy for readers to look at the wrong legend.

• As previously noted, consider further paneling violin plots per result topic or moving some of these figures to the supplement.

• Table 1 – define what “trigger” means in the model context either in the table caption or somewhere in the text.

• Table 4 could also be moved to the supplement as it is not strictly necessary for interpreting the modeling process, in my opinion.

**Do you want your identity to be public for this peer review?** For information about this choice, including consent withdrawal, please see our Privacy Policy

Reviewer #1: No

Reviewer #2: No

---

## [Author Response · Author response to Decision Letter 1]

30 Jul 2025

Please see the attached file (Response to Reviewers.docx) for all responses to reviewer comments - the responses do not format well here.

---

## [Decision Letter · Decision Letter 1]

19 Aug 2025

Dear Dr. Busby,

Thank you for submitting your manuscript to PLOS ONE. After careful consideration, we feel that it has merit but does not fully meet PLOS ONE’s publication criteria as it currently stands. Therefore, we invite you to submit a revised version of the manuscript that addresses the points raised during the review process.

We look forward to receiving your revised manuscript.

Kind regards,

Rajan Parajuli

Academic Editor

PLOS ONE

Journal Requirements:

Additional Editor Comments:

All three reviewers found that authors addressed the comments and suggestions in this version, however, before it gets published, authors should consider fixing the minor comments from Reviewers 1 and 3.

Reviewers' comments:

Reviewer's Responses to Questions

**Comments to the Author**

Reviewer #1: (No Response)

Reviewer #2: All comments have been addressed

Reviewer #3: All comments have been addressed

2. Is the manuscript technically sound, and do the data support the conclusions?

Reviewer #1: Yes

Reviewer #2: Yes

Reviewer #3: Yes

3. Has the statistical analysis been performed appropriately and rigorously?

Reviewer #1: Yes

Reviewer #2: Yes

Reviewer #3: Yes

4. Have the authors made all data underlying the findings in their manuscript fully available?

Reviewer #1: Yes

Reviewer #2: Yes

Reviewer #3: No

5. Is the manuscript presented in an intelligible fashion and written in standard English?

Reviewer #1: Yes

Reviewer #2: Yes

Reviewer #3: Yes

Reviewer #1: I commend the authors for making a number of significant changes to the manuscript in response to the reviewers' comments on the initial submission. The addition of more nuanced descriptions and discussion of common silvicultural treatment regimes and implications of the paper's results across different ownerships in the introduction and discussion, the addition of relevant statistical analyses to their tradeoffs assessment, and the overall streamlining and re-focusing of the discussion have greatly improved the manuscript. This revised manuscript offers compelling insights and consideration of what “fire-aware” management might look like in moist, westside PNW forests, and what the implications are for a variety of resource values. I have only a few minor suggested to consider before the eventual publication of the manuscript.

L426-435: This interpretation and explanation of the results would be more appropriate for the discussion.

L586-593: I applaud the author’s addition of appropriate statistical tests for their tradeoff analyses, but some version of this explanation of how they treated and analyzed the data should be incorporated into the methods section.

Reviewer #2: (No Response)

Reviewer #3: I found that the authors have addressed the comments very well and substantially revised the manuscript, and it is acceptable for publication. However, I have some concerns

Some Acronyms, such as FF, FFCA, Rx, and others, have been repeatedly spelled out throughout the paragraph. Is this a common standard or something else? Please fix this issue.

I suggest using the full y-axis range while showing the range of distribution, for example, Fig. 7, where the upper and lower quantile range is cut off in some violin, due to the constrained y-axis range. Check the similar issue with Fig. 9, and S3, S5, S7

Fig 8- provide a negative y-axis value for the 6th right panel (Washington); as well Fig S4.

**Do you want your identity to be public for this peer review?** For information about this choice, including consent withdrawal, please see our Privacy Policy

Reviewer #1: No

Reviewer #2: No

Reviewer #3: No

---

## [Author Response · Author response to Decision Letter 2]

25 Aug 2025

Please see the "Response to Reviewers" document attached to the manuscript package.

---

## [Editor Report · Decision Letter 2]

27 Aug 2025

Prospects for silvicultural enhancement of fire resistance in mesic westside forests of the Pacific Northwest.

PONE-D-25-22508R2

Dear Dr. Busby,

We’re pleased to inform you that your manuscript has been judged scientifically suitable for publication and will be formally accepted for publication once it meets all outstanding technical requirements.

Kind regards,

Rajan Parajuli

Academic Editor

PLOS ONE
---

## [Editor Report · Acceptance letter]

PONE-D-25-22508R2

PLOS ONE

Dear Dr. Busby,

I'm pleased to inform you that your manuscript has been deemed suitable for publication in PLOS ONE. Congratulations! Your manuscript is now being handed over to our production team.

Kind regards,

on behalf of

Dr. Rajan Parajuli

Academic Editor

PLOS ONE